# DATAGEN: UNIFIED SYNTHETIC DATASET VIA LARGE LANGUAGE MODELS

**Yue Huang[1,†], Siyuan Wu[2,†], Chujie Gao[3], Dongping Chen[2,4], Qihui Zhang[5], Yao Wan[2,*]**
**Tianyi Zhou[6], Chaowei Xiao[7], Jianfeng Gao[8], Lichao Sun[9,*], Xiangliang Zhang[1,*]**

[1]University of Notre Dame, [2]Huazhong University of Science and Technology, [3]MBZUAI
[4]University of Washington, [5]Peking University, [6]University of Maryland, College Park
[7]University of Wisconsin–Madison, [8]Microsoft Research, [9]Lehigh University

## ABSTRACT

Large Language Models (LLMs) such as GPT-4 and Llama3 have significantly impacted various fields by enabling high-quality synthetic data generation and reducing dependence on expensive human-generated datasets. Despite this, challenges remain in the areas of generalization, controllability, diversity, and truthfulness within the existing generative frameworks. To address these challenges, this paper presents DATAGEN, a comprehensive LLM-powered framework designed to produce diverse, accurate, and highly controllable datasets. DATAGEN is adaptable, supporting all types of text datasets and enhancing the generative process through innovative mechanisms. To augment data diversity, DATAGEN incorporates an attribute-guided generation module and a group checking feature. For accuracy, it employs a code-based mathematical assessment for label verification alongside a *retrieval-augmented generation* technique for factual validation. The framework also allows for user-specified constraints, enabling customization of the data generation process to suit particular requirements. Extensive experiments demonstrate the superior quality of data generated by DATAGEN, and each module within DATAGEN plays a critical role in this enhancement. Additionally, DATAGEN is applied in two practical scenarios: benchmarking LLMs and data augmentation. The results indicate that DATAGEN effectively supports dynamic and evolving benchmarking and that data augmentation improves LLM capabilities in various domains, including agent-oriented abilities and reasoning skills.

## 1 INTRODUCTION

Large Language Models (LLMs) such as GPT-4 (OpenAI, 2023a), Claude (Anthropic, 2023), and Llama3 (Meta, 2023) have demonstrated excellent performance across various professional domains, including medical (Liu et al., 2023a; Zhang et al., 2024a), educational (Kasneci et al., 2023), software engineering (Qian et al., 2023), and social sciences (Li et al., 2024a;b), as well as in LLM-based agent applications (Huang et al., 2023a; Liu et al., 2023b; Chen et al., 2024a). Given their superior generative capabilities, it is natural for researchers to explore effective methods for utilizing these models in synthetic data generation (Zhu et al., 2024a;b; Wang et al., 2024a). The primary goal is to produce high-quality, cost-effective datasets, thereby reducing the reliance on expensive human labor. Furthermore, data generated by LLMs can be utilized for data augmentation (Yu et al., 2024), dynamic evaluation (Zhu et al., 2024a;b), and model self-alignment (Sun et al., 2023).

Despite the advancements in LLM-driven data generation (Zhu et al., 2024a;b; Wang et al., 2024a; Dekoninck et al., 2024a;b), which have significantly improved the data generation pipeline and reduced the human cost, some challenges remain: **(1) Generalization and Controllability:** Most of existing frameworks directly modify data items in original datasets in specific ways based on fixed principles (Zhu et al., 2024b; Wang et al., 2024a) (*e.g.*, add additional context or shuffle the order of

---

*Corresponding Authors.
†Yue and Siyuan contributed equally to this work.

Table 1: Comparison of different dataset generation frameworks. The gray checkmark means the work may achieve parts of the goal (not all).

| Related Work | General-ization | Control-lability | Diversity | Truthful-ness | *w/o* Human Intervention | New Knowledge | Dynamic Benchmark | Data Aug. |
|---|---|---|---|---|---|---|---|---|
| DyVal (Zhu et al., 2024a) | ✗ | ✗ | ✗ | ✓ | ✓ | ✗ | ✓ | ✓ |
| DyVal 2 (Zhu et al., 2024b) | ✓ | ✓(gray) | ✗ | ✗ | ✓ | ✓(gray) | ✓ | ✓ |
| S3Eval (Lei et al., 2024) | ✗ | ✓ | ✗ | ✗ | ✓ | ✗ | ✓ | ✗ |
| Yu et al. (2024) | ✓(gray) | ✓ | ✓ | ✗ | ✓ | ✓ | ✗ | ✓ |
| Chung et al. (2023) | ✗ | ✗ | ✓ | ✓ | ✗ | ✗ | ✗ | ✗ |
| Fan et al. (2024) | ✗ | ✗ | ✗ | ✓ | ✓ | ✗ | ✓ | ✗ |
| Jandaghi et al. (2023) | ✗ | ✗ | ✗ | ✗ | ✓ | ✓ | ✗ | ✗ |
| Wang et al. (2024a) | ✓ | ✗ | ✗ | ✓(gray) | ✓ | ✓(gray) | ✓ | ✗ |
| MetaMath (Yu et al., 2023) | ✗ | ✗ | ✓ | ✗ | ✓ | ✗ | ✗ | ✓ |
| Qameleon (Agrawal et al., 2023) | ✗ | ✗ | ✗ | ✗ | ✗ | ✗ | ✗ | ✓ |
| Viswanathan et al. (2023) | ✗ | ✗ | ✓ | ✗ | ✓ | ✓ | ✗ | ✓ |
| Chen et al. (2024b) | ✗ | ✗ | ✓ | ✓ | ✓ | ✓ | ✗ | ✓ |
| Gandhi et al. (2024) | ✗ | ✗ | ✓ | ✗ | ✓ | ✗ | ✗ | ✗ |
| **DATAGEN (Ours)** | ✓ | ✓ | ✓ | ✓ | ✓ | ✓ | ✓ | ✓ |

the options), which may constrain the generalization of the generated data as they do not modify the nature of the data items like the scenarios within items. Moreover, many of them are also limited to particular dataset formats or types (Yu et al., 2024; Zhu et al., 2024a), such as multiple-choice or mathematically-oriented datasets (*e.g.*, GSM8K (Cobbe et al., 2021)). Additionally, the lack of provisions for incorporating external constraints, like specific user requirements (*e.g.*, users may specify the length of generated text), restricts their controllability during generation. **(2) Diversity and Truthfulness:** Prior efforts always overlook the need to ensure some quality aspects of the datasets like diversity and truthfulness. For instance, the direct application of LLMs for dataset generation often leads to replication and low diversity, as LLMs may output the same answers when faced with semantically similar input. Furthermore, the propensity of LLMs to produce hallucinations (Huang et al., 2023b; Sun et al., 2024a) can introduce factual inaccuracies, potentially degrading model performance when such datasets are used for training or fine-tuning.

To address these challenges, this paper puts forward DATA-GEN (as shown in Figure 1), a unified and LLM-powered framework designed to generate a dataset. DATAGEN ensures the generalization, diversity, truthfulness, and controllability simultaneously of the generation process, compared to previous studies (as shown in Table 1). DATAGEN accepts all kinds of text datasets and generates high-quality datasets based on various modules. To enrich the diversity of the generated datasets, DATAGEN employs a range of strategies, including various hyperparameter settings, attribute-guided generation, and group checking. To guarantee the truthfulness of the generated datasets, we propose a code-based mathematical assessment to detect and rectify potentially incorrect labels. Additionally, we adopt a Retrieval-Augmented Generation (RAG)-based validation method to check the factuality of generated statements to ensure their truthfulness. DATAGEN integrates constraints input to align with user specifications to enhance user control over the dataset generation process. Furthermore, by employing attribute-guided generation and difficulty enhancement, we enable the generation of data covering a wide range of topics while providing users with controllable difficulty levels.

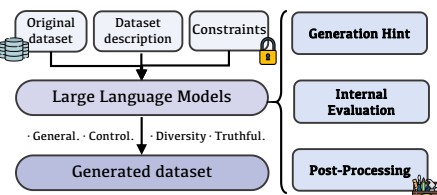

Figure 1: Our proposed DATAGEN for dataset generation via LLMs.

To summarize, the key contributions of this paper are as follows:

- We introduce DATAGEN, a unified framework for generating textual datasets via LLMs, which accepts the original dataset, description, and user constraints, and integrates modules to ensure diversity, truthfulness, and controllability.
- We extensively evaluate DATAGEN across data characterization, module efficacy, human evaluation, error analysis, and cost analysis, confirming its proficiency in dataset generation and highlighting promising future research directions.

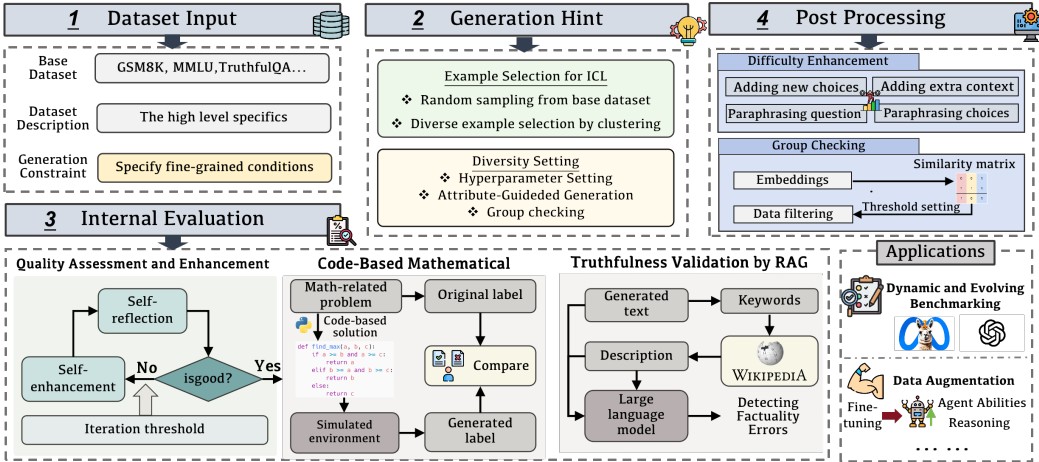

Figure 2: The architecture of DATAGEN.

- We explore two applications of DATAGEN: benchmarking LLMs and data augmentation. Key insights include: I) Most LLMs struggle with math-oriented datasets generated by DATAGEN (*e.g.*, GSM8K). II) The performance of LLMs varies significantly across datasets generated by different LLMs. III) LLMs' capabilities across various aspects (*e.g.*, agent-related abilities, reasoning skills) can be improved by fine-tuning based on the generated data. IV) An improvement of data augmentation exists in knowledge-intensive datasets.

## 2 DATAGEN FRAMEWORK

In this section, we will introduce the proposed DATAGEN, a unified framework for dataset generation. DATAGEN consists of four modules (as shown in Figure 2) including framework input, generation hint, internal evaluation, and post-processing. Formally, consider an original dataset $\mathcal{D}$, the proposed framework $\mathcal{F}$, which operates by iteratively sampling subsets $\mathcal{S}_i$ from $\mathcal{D}$ (*i.e.*, example selection for few-shot learning in Section 2.2). For each subset, $\mathcal{F}$ applies transformations based on the dataset's description $\mathcal{M}(\mathcal{D})$ and a set of constraints $\mathcal{C}$. The final generated dataset, $\mathcal{D}_{\text{gen}}$, is accumulated over $N$ iterations: $\mathcal{D}_{\text{gen}} = \bigcup_{i=1}^{N} \mathcal{F}(\mathcal{S}_i, \mathcal{M}(\mathcal{D}), \mathcal{C})$. During generation, the objectives of DATAGEN focus on maximizing the generalization, controllability, diversity, and truthfulness of the generated dataset.

### 2.1 FRAMEWORK INPUT

The input for DATAGEN comprises three components: *base dataset*, *dataset description*, and *generation constraints*: The *base dataset* is provided in a standardized JSON format, which may include text with a label or standalone text (*e.g.*, "text with a label" or "single text"). The *dataset description* articulates the specifics of the base dataset at a high level, furnishing foundational guidance for the LLM to synthesize a dataset analogous to the original. While optional, the *generation constraints* (Zhou et al., 2023) specify fine-grained conditions under which the LLM operates during dataset generation. For instance, constraints might stipulate that "*Do not generate text longer than 20 words*" or "*Include an emoji in each generated sample*", thereby restricting specific conditions of the synthetic dataset.

### 2.2 GENERATION HINT

**Few-Shot Learning.** The base dataset typically comprises hundreds of data items; however, incorporating all these items directly into the prompt may result in an excessively long context that could obscure the comprehension capabilities of LLMs and incur substantial costs (Bai et al., 2023). To mitigate these challenges, few-shot learning techniques are employed for dataset generation (Brown et al., 2020; Wang et al., 2020). Within DATAGEN, two principal methods are utilized to select few-shot learning examples. The first method involves a random sampling from the base dataset, effectively reducing both generation time and associated costs. The second method focuses on enhancing the diversity of examples, thereby guiding LLMs to generate as varied a dataset as possible. Specifically,

DATAGEN initially encodes all data items using OpenAI's `text-embedding-ada-002` (OpenAI, a) to create an embedding list. Subsequently, a clustering algorithm (e.g., K-means (Hartigan and Wong, 1979)) is applied to form $n$ clusters, where $n$ represents the desired number of examples. One example is randomly selected from each cluster, yielding a set of $n$ diverse examples.

**Diversity Setting.** To augment the diversity of the generated data, we implement two strategies: (1) *Hyperparameter Setting.* The content generated by LLMs is influenced by various factors, with hyperparameters such as temperature, top-k, and top-p being crucial. To maximize the diversity of the dataset, we manipulate these hyperparameters, particularly the temperature settings. (2) *Attribute-Guided Generation.* Drawing on insights from prior research (Huang et al., 2023a; Yu et al., 2024), we formalize the attribute-guided text generation process for LLMs. Let $\mathcal{A} = \{a_1, a_2, \ldots, a_n\}$ be a set of attributes, such as "economics" and "sports", intended to guide the generation process. We model the generation process as a function where the output text $y$ is a function of the input prompt $x$ and a vector of attributes $\mathbf{a} \in \mathcal{A}$. The generation process can be expressed as $y = P(x, \mathbf{a})$, where $P$ represents the generation model of the LLM, and $x$ is the input prompt. To implement this, we employ two distinct strategies: the first involves directly incorporating user-input customized attributes, and the second requires asking LLMs to extract necessary attributes from given data examples (the prompt template is shown in Appendix I). (3) *Group Checking.* To ensure diversity among the generated items, a similarity matrix is employed to identify and filter out pairs of data items exhibiting high similarity. Further details on this process are provided in Section 2.4.

## 2.3 INTERNAL EVALUATION

**Overall Quality Assessment and Enhancement.** After obtaining the raw generated data, it's important to enhance their overall quality as during the generation, LLMs may overlook some details so as to mistake like deviating from the dataset description. Inspired by recent studies about self-evaluation and self-alignment (Ji et al., 2023; Ren et al., 2023; Huang et al., 2023c; Jain et al., 2023; Sun et al., 2023; Wang et al., 2023), we leverage LLMs themselves to improve the quality of generated data. The process involves two primary steps: (1) *Self-Reflection.* Each generated data item is initially subjected to a self-reflection phase, wherein LLMs assess the item to determine errors and potential areas for enhancement. The output of self-reflection contains two parts: whether the given data needs to be enhanced and the reason why it needs enhancement. (2) *Self-Enhancement.* When LLMs recognize the necessity for improvements, both the reflective insights and the data item itself are re-input into the LLM to generate an improved version. By establishing a threshold for the number of iterations and repetitively applying these steps, DATAGEN effectively elevates the overall quality of the generated items.

**Code-Based Mathematical Evaluation.** In generating mathematics-related datasets, such as GSM8K (Cobbe et al., 2021), it has been observed that a proportion of generated labels are factually incorrect. To address this issue, we employ a code-based mathematical evaluation method to verify the accuracy of generated labels. As highlighted in the recent study by (Gou et al., 2024; Chen et al., 2023), the use of tools (*e.g.*, a Python function) can substantially improve reasoning performance. Motivated by this finding, we require the LLM to generate Python code to solve the given math-related problem. The code is then executed within a simulative environment to produce a solution. The code-verified answer(*i.e.*, label) is subsequently compared with the original LLM-generated answer. If they conflict, the original LLM-generated answer will be replaced with the code-verified answer.

**Truthfulness Validation by RAG.** Ensuring the truthfulness of generated golden answers is crucial when creating datasets that require factual knowledge. Prior studies have utilized Retrieval-Augmented Generation (RAG) to enhance the factuality and reduce the incidence of hallucinations in LLMs (Aksitov et al., 2023; Li et al., 2024c; 2022; Gao et al., 2024a). To combat hallucinations within the generated data, we implement a RAG-based validation process in DATAGEN. Specifically, the LLM first identifies keywords from the generated text. Subsequently, DATAGEN retrieves relevant descriptions based on these keywords from the Wikipedia database, as demonstrated in prior research (Semnani et al., 2023). These descriptions are then used as prompts to guide the LLM in detecting and correcting any discrepancies or errors in the generated content.

## 2.4 POST-PROCESSING

**Difficulty Enhancement.** Given that the dataset is produced by LLMs, the complexity of the generated data is occasionally insufficient to challenge LLMs as their capabilities evolve. To address

this, and inspired by prior research (Wang et al., 2024a; Zhu et al., 2024b), we implement several strategies to increase the data's difficulty. These strategies are designed to elevate the challenges faced by LLMs in processing and responding to the data. The applied policies include: (1) *Paraphrasing Question:* Reformulate the phrasing to express the same idea with greater sophistication. (2) *Adding Extra Context into Question:* Integrate additional context or details that, while not directly aiding in the question's resolution, enhance the question's complexity. (3) *Paraphrasing The Choices:* Each option should be rephrased to reflect the same concept or idea as the original. The essence and meaning must be preserved. If an option cannot be paraphrased without altering its meaning, it should remain unchanged. (4) *Adding A New Choice:* Introduce a plausible but incorrect option to the existing choices to create ambiguity and require deeper understanding.

**Group Checking.** To mitigate the issue of high similarity among generated data items, a group-checking mechanism is implemented to identify and eliminate duplicates. Specifically, we utilize OpenAI's `text-embedding-ada-002` (OpenAI, a) to compute embeddings for all generated items. Let $\mathcal{X} = \{x_1, x_2, \ldots, x_n\}$ be the set of generated data items, and $\mathbf{e}_i$ be the embedding of item $x_i$ computed via `text-embedding-ada-002`. We define the similarity matrix $\mathbf{S}$ where the element $s_{ij}$ is given by $s_{ij} = \sqrt{\sum_{k=1}^{d}(e_{ik} - e_{jk})^2}$, representing the Euclidean distance between the embeddings of items $x_i$ and $x_j$. Data items exhibiting a similarity exceeding a predefined threshold $\theta$ are filtered out to ensure diversity within the dataset. Formally, if $s_{ij} < \theta$ for any pair $(i, j)$, at least one of the items $x_i$ or $x_j$ is randomly removed from the final dataset.

## 3 EXPERIMENTS AND APPLICATIONS

### 3.1 EXPERIMENTAL SETUP

| Type | GSM8K | MMLU | TruthfulQA | HellaSwag |
|------|-------|------|------------|-----------|
| **Generated** | 0.663 | 0.744 | 0.743 | 0.680 |
| **Original** | 0.681 | 0.746 | 0.745 | 0.742 |
| $\Delta$ | 2.64% | 0.27% | 0.27% | 8.36% |

Table 2: Remote-Clique of generated data and original data. $\Delta$ is the difference between them.

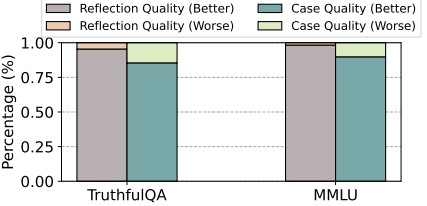

Figure 3: Human evaluation of overall quality assessment and enhancement.

To thoroughly evaluate the effectiveness of DATAGEN, we carefully select four representative benchmark datasets: GSM8K (Cobbe et al., 2021), TruthfulQA (Lin et al., 2022), MMLU (Hendrycks et al., 2021a), and HellaSwag (Zellers et al., 2019). Each dataset uniquely contributes to language model assessment, covering dimensions from mathematical problem-solving and factual accuracy verification to extensive language understanding and commonsense reasoning. We show the details of these four datasets in Appendix C. For dataset generation, we utilize GPT-4 (OpenAI, 2023a), Claude3-Opus (Anthropic, 2023), and Llama3-70b (Meta, 2023), as these LLMs are among the most robust available, exhibiting exceptional ability to follow instructions. For benchmarking, our study utilizes eight popular models from notable entities in the AI domain (the details are shown in Appendix C.), reflecting a mix of open-source and proprietary technologies. The number of generated data items and more details are shown in Appendix E. Note that difficulty enhancement is not applied to the generated data for benchmarking. We will discuss the effectiveness of difficult enhancement in Section 3.3. All LLMs utilized for generation share the same prompt templates.

### 3.2 CHARACTERIZING GENERATED DATA

**Length.** As depicted in Figure 4a, the length distribution of all generated datasets approximates a normal distribution. Notably, except for the HellaSwag dataset (as the length of the original HellaSwag dataset looks like a bimodal distribution), the distributions of other datasets closely resemble those of their original datasets. This similarity indicates that DATAGEN effectively mimics the distribution of the original data, thereby enhancing the reliability of the generated datasets.

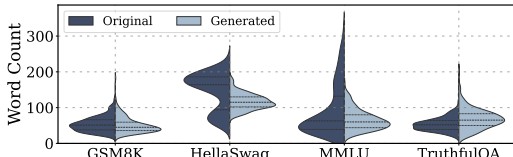
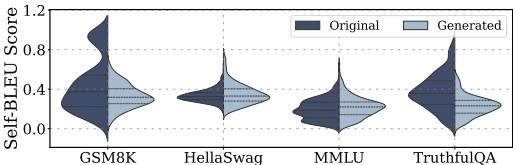

(a) Len. distribution of generated and original data.    (b) Self-BLEU of generated and original data.

Figure 4: Length and the self-BLEU score of generated data and original data.

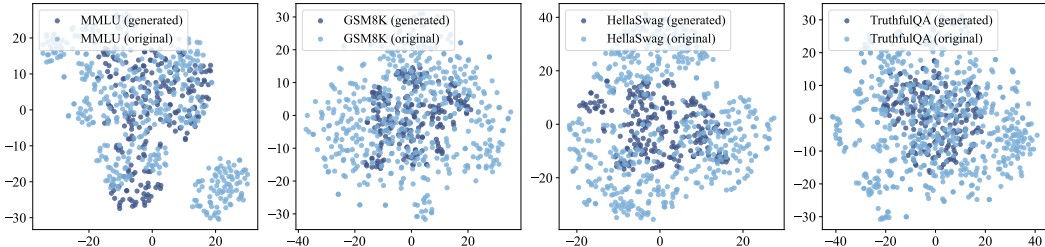

Figure 5: Semantic embedding of different datasets. We use OpenAI's `text-embedding-ada-002` (OpenAI, a) to obtain text embedding.

**Semantic Embedding.** As illustrated in Figure 5, the distribution of the generated dataset is encompassed within the distribution of the original dataset. This observation indicates that the data items generated are semantically aligned with the original data, confirming their semantic correctness.

**Diversity.** Analogous to the length distribution, the distribution of the self-BLEU score (Zhu et al., 2018) (as depicted in Figure 4b)—a metric employed to assess text diversity—indicates that the diversity of the generated data closely aligns with that of the original dataset. This alignment underscores the exceptional capability of DATAGEN to replicate the diversity inherent in the original dataset, demonstrating its effectiveness in producing varied textual content. Additionally, we utilize the remote-clique metric, as applied in prior research (Cevallos et al., 2018), to measure the diversity of the generated data. The related statistics are presented in Table 2. Observations reveal that the remote-clique scores of the original and generated data are closely matched, with less than 10% variance, affirming that our generated data maintains a high level of diversity comparable to the original dataset.

**Knowledge Richness Introduced.** In contrast to prior research (Zhu et al., 2024a;b; Wang et al., 2024a), DATAGEN innovates by generating entirely new data items, rather than merely modifying existing answers. This approach introduces novel scenarios and knowledge. We assess the knowledge richness of the data generated by DATAGEN and compared it to the previous study (i.e., Dyval2 (Zhu et al., 2024b)) by calculating the entity overlap rate—how many entities appear both in the generated and original data. A lower overlap rate indicates that the framework is introducing more new knowledge. According to our findings, presented in Table 3, DATAGEN demonstrates an average overlap rate of only 3.83%, significantly lower than that of Dyval2 (Zhu et al., 2024b). This substantial reduction in overlap rate signifies that our framework excels at incorporating new knowledge into the generated datasets.

**Influence of temperature.** We examine the impact of temperature settings on the diversity of data generated by GPT-4. For this purpose, we select a few items from the TruthfulQA dataset to use as examples in few-shot learning. We conduct experiments using temperature settings of 0 and 1. Our findings indicate that the Remote-Clique score (Cevallos et al., 2018) at a temperature of 0 is 0.683, whereas, at a temperature of 1, it increases to 0.721. This suggests that adjusting the temperature setting can significantly enhance the diversity of the generated data.

### 3.3    EFFECTIVENESS OF MODULES IN DATAGEN

In this section, we validate the effectiveness of modules in DATAGEN. To simplify the analysis, our evaluation is based on the GPT-4 generated data: **(1) Diversity Setting.** As demonstrated in Table 4,

Table 3: The knowledge richness comparison between different principles in DyVal 2 (Zhu et al., 2024b) and DATAGEN. The principle 1, 2, 3, and 4 are paraphrasing questions, paraphrasing choices, adding extra context to questions, and adding a new choice.

| Baseline | HellaSwag | MMLU | TruthfulQA | Avg. |
|---|---|---|---|---|
| **DyVal2-prin.1** | 24.30% | 61.30% | 51.40% | 45.67% |
| **DyVal2-prin.2** | 40.50% | 65.70% | 46.20% | 50.80% |
| **DyVal2-prin.3** | 27.00% | 62.70% | 57.30% | 49.00% |
| **DyVal2-prin.4** | 51.40% | 71.00% | 47.60% | 56.67% |
| **DATAGEN** | **5.40%** | **3.30%** | **2.80%** | **3.83%** |

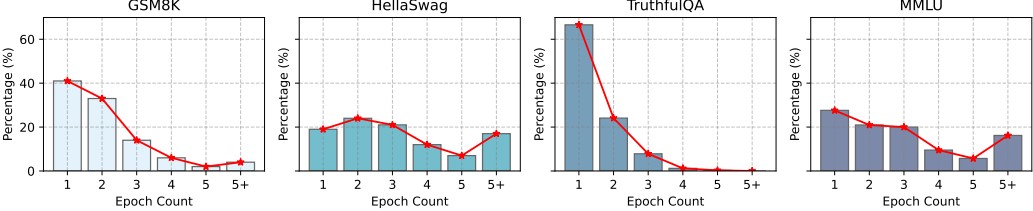

Figure 6: The percentage of different epoch counts in four datasets.

the DATAGEN modules significantly enhance the diversity of the generated data. Specifically, the remote-clique score of the initially generated data stands at 0.695. However, the introduction of attribute-guided generation elevates the remote-clique score to 0.735. Furthermore, the implementation of group checking further increases this score to 0.743. **(2) Overall Quality Assessment and Enhancement.** To evaluate the effectiveness of our quality assessment and enhancement module, we conducted human evaluations focusing on two key aspects: (I) Comparing the quality between original and enhanced data items; (II) Assessing the reasonableness of the reflections. As illustrated in Figure 3, the results indicate that almost all reflections were deemed reasonable by the evaluators. Furthermore, over 80% of the enhanced data items were rated as superior in both datasets. These findings underscore the effectiveness of our module. **(3) Difficulty Enhancement.** As demonstrated in Table 6, it is observable that the performance of most of the LLMs declined when compared to their performance on the baseline-generated datasets after the application of difficulty enhancement. This result underscores the effectiveness of difficulty enhancement, which suggests its potential utility in preventing data contamination (Dong et al., 2024; Golchin and Surdeanu, 2024; Xu et al., 2024a). Such techniques may thus contribute significantly to improving the robustness of LLMs against overfitting to training datasets. **(4) Code-Based Mathematical Evaluation.** As depicted in Table 4, our code-based evaluation methodology has significantly enhanced the correctness of the generated data, improving from an initial accuracy of 44% to 92%. **(5) Truthfulness Validation by RAG.** As detailed in Table 4, the RAG-based validation corrected 4.2% of the examples, demonstrating its effectiveness. This percentage also highlights the high quality of the dataset generated by GPT-4, which contains only a few errors. The correctness of (4) and (5) are also manually evaluated, which of the details can be found in Appendix D.

In Appendix F, we investigate the impact of temperature settings on data diversity and evaluate the adherence of LLMs in DATAGEN to user constraints. Our findings reveal that adjusting the temperature setting enhances the diversity of generated data. Furthermore, LLMs within DATAGEN effectively follow user-imposed constraints in both individual and combined scenarios. We also provide a cost analysis of DATAGEN in Appendix F, demonstrating that DATAGEN generates datasets at a significantly low cost.

Table 4: Effectiveness of each module in DATAGEN.

| Diversity Enhancement | | | Code-based. | | RAG Validation |
|---|---|---|---|---|---|
| Raw | +Attribute Guided | +Group Checking | Raw | +Validation | Corrected Percentage |
| 0.695 | 0.735 (5.8% ↑) | 0.743 (6.9% ↑) | 44% | 88% | 4.2% |

| Error Type | GSM8K | HellaSwag | MMLU | TruthfulQA |
|---|---|---|---|---|
| **Factuality Error** | 41% | 14% | 69% | 79% |
| **Format Error** | 20% | 29% | 8% | 0% |
| **Multiple Answers** | 0% | 43% | 0% | 0% |
| **Question Error** | 39% | 14% | 23% | 21% |

Table 5: Proportion of different errors. Multiple answers mean the question is considered to have multiple correct answers after human evaluation. Question errors mean the question has quality flaws like unclear statements.

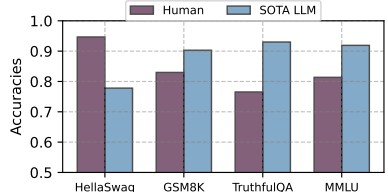

Figure 7: Performance of human and the best LLM (SOTA LLM) on four generated datasets.

## 3.4 HUMAN PERFORMANCE ON GENERATED DATASET

As depicted in Figure 7, the performance comparison between humans and LLMs reveals distinct outcomes across various datasets. In the HellaSwag dataset, human performance slightly surpasses that of LLMs. However, in the other three datasets, LLMs demonstrate superior performance. Notably, in the GSM8K dataset, the accuracy of human responses is lower than that of the best-performing LLM. For the TruthfulQA and MMLU datasets, which require extensive knowledge, humans perform significantly worse than LLMs, which benefit from training on large, diverse corpora. More details about evaluating human performance are shown in Appendix D.

## 3.5 ERROR ANALYSIS

To examine the errors present in the generated dataset, we conducted a human evaluation for error analysis. We observe significant factuality errors in datasets such as GSM8K, TruthfulQA, and MMLU, primarily because these datasets contain responses that are fact-based (*e.g.*, arithmetic question answers). This observation underscores the necessity for enhancements in the accuracy of provided answers. Despite the robust instruction-following capabilities of GPT-4, it occasionally struggles with data formatting issues. Such errors could be mitigated through clearer prompts or by employing an integrated framework like LangChain[*]. Additionally, our analysis of the HellaSwag dataset revealed the presence of multiple viable answers for certain prompts, highlighting the need for a more comprehensive answer validation mechanism. We discuss the potential improvement by mitigating these errors in Appendix A.

## 3.6 COST ABLATION ANALYSIS

We conduct a cost analysis of DATAGEN. Specifically, we calculate the total token usage and the corresponding cost for generating data across four datasets: MMLU, HellaSwag, TruthfulQA, and GSM8K. The details are presented in Figure 8.

For a generated item without RAG-based validation and code-based evaluation, the cost is at most $0.038 using the GPT-4-Turbo API. When incorporating RAG-based validation, the average cost per generated item increases to $0.190, due to the large volume of tokens processed from the retrieved content. Adding code-based evaluation raises the cost to $0.040. Overall, the total cost for generating each item, including all validation and evaluation processes, will not exceed $0.200. This cost, although significant, is substantially lower than the cost of human labor.

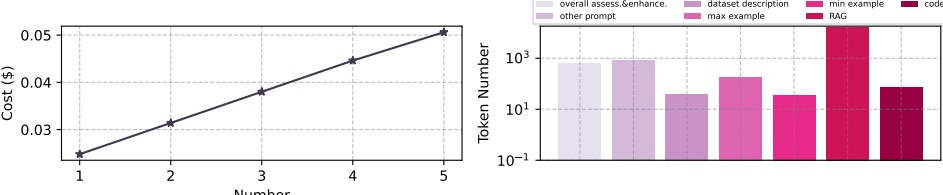

Figure 8: Cost (dollar) on different epoch numbers of overall quality assessment and enhancement (Left), and the token number cost of each part in DATAGEN.

---

[*]https://github.com/langchain-ai/langchain

Table 6: LLMs' performance on baseline generated (*i.e.*, *gen.*) dataset, challenge or difficulty enhanced dataset (*i.e.*, *cha.*), and their differences (*i.e.*, *diff.*).

| Model | GSM8K | | | MMLU | | | HellaSwag | | | TruthfulQA | | |
|---|---|---|---|---|---|---|---|---|---|---|---|---|
| | *gen.* | *cha.* | *diff.* | *gen.* | *cha.* | *diff.* | *gen.* | *cha.* | *diff.* | *gen.* | *cha.* | *diff.* |
| **ChatGPT** | 0.665 | 0.585 | 0.080 | 0.798 | 0.633 | 0.165 | 0.960 | 0.924 | 0.036 | 0.816 | 0.718 | 0.098 |
| **Claude-3** | 0.778 | 0.670 | 0.108 | 0.903 | 0.725 | 0.178 | 0.935 | 0.880 | 0.055 | 0.919 | 0.810 | 0.109 |
| **Llama3-70b** | 0.689 | 0.637 | 0.052 | 0.857 | 0.703 | 0.154 | 0.949 | 0.884 | 0.065 | 0.914 | 0.743 | 0.171 |
| **Llama3-8b** | 0.613 | 0.557 | 0.056 | 0.741 | 0.576 | 0.165 | 0.793 | 0.699 | 0.094 | 0.795 | 0.676 | 0.119 |
| **Mistral-7b** | 0.377 | 0.321 | 0.056 | 0.709 | 0.437 | 0.272 | 0.696 | 0.467 | 0.229 | 0.738 | 0.452 | 0.286 |
| **Mixtral-8x7b** | 0.509 | 0.439 | 0.070 | 0.851 | 0.616 | 0.235 | 0.511 | 0.373 | 0.138 | 0.824 | 0.648 | 0.176 |
| **Yi-34b** | 0.637 | 0.509 | 0.128 | 0.815 | 0.633 | 0.182 | 0.572 | 0.522 | 0.050 | 0.857 | 0.657 | 0.200 |

Table 7: The main results on generated datasets (*i.e.*, *gen.*) and original datasets (*i.e.*, *ori.*).

| Dataset | GSM8K | | MMLU | | TruthfulQA | | HellaSwag | |
|---|---|---|---|---|---|---|---|---|
| | *ori.* | *gen.* | *ori.* | *gen.* | *ori.* | *gen.* | *ori.* | *gen.* |
| *GPT-4 Generation* | | | | | | | | |
| **ChatGPT** | 0.762 | 0.665 | 0.609 | 0.798 | 0.825 | 0.837 | 0.611 | 0.960 |
| **Claude-3** | 0.953 | 0.778 | 0.810 | 0.903 | 0.855 | 0.919 | 0.888 | 0.935 |
| **Llama3-70b** | 0.890 | 0.689 | 0.755 | 0.857 | 0.750 | 0.914 | 0.836 | 0.949 |
| **Llama3-8b** | 0.800 | 0.613 | 0.565 | 0.741 | 0.450 | 0.795 | 0.684 | 0.793 |
| **Mistral-7b** | 0.313 | 0.377 | 0.490 | 0.709 | 0.382 | 0.738 | 0.600 | 0.696 |
| **Mixtral-8x7b** | 0.610 | 0.509 | 0.720 | 0.851 | 0.640 | 0.824 | 0.712 | 0.511 |
| **Yi-34b** | 0.687 | 0.637 | 0.645 | 0.815 | 0.485 | 0.857 | 0.740 | 0.572 |
| *Claude-3-Opus Generation* | | | | | | | | |
| **ChatGPT** | 0.762 | 0.405 | 0.609 | 0.802 | 0.432 | 0.744 | 0.538 | 0.712 |
| **GPT-4** | 0.947 | 0.508 | 0.725 | 0.848 | 0.841 | 0.888 | 0.736 | 0.835 |
| **Llama3-70b** | 0.890 | 0.444 | 0.755 | 0.846 | 0.750 | 0.854 | 0.836 | 0.769 |
| **Llama3-8b** | 0.800 | 0.367 | 0.565 | 0.780 | 0.450 | 0.709 | 0.568 | 0.704 |
| **Mistral-7b** | 0.313 | 0.158 | 0.490 | 0.709 | 0.380 | 0.621 | 0.580 | 0.690 |
| **Mixtral-8x7b** | 0.610 | 0.291 | 0.720 | 0.717 | 0.640 | 0.680 | 0.600 | 0.565 |
| **Yi-34b** | 0.687 | 0.323 | 0.645 | 0.751 | 0.480 | 0.694 | 0.644 | 0.584 |

## 3.7 APPLICATION-I: BENCHMARKING LLMS

We present the benchmarking results based on GPT-4 and Claude3 generated data for seven popular LLMs in Table 7 (the benchmarking results based on Llama3-70b's generation are shown in Appendix F). The analysis yields several key observations:

- **Performance decline on generated GSM8K dataset:** Almost all LLMs exhibit a performance drop on the generated GSM8K dataset compared to the original. This suggests that the reasoning capabilities of many LLMs may be overstated, aligning with recent findings (Zhang et al., 2024b; Mirzadeh et al., 2024; Zhang et al., 2024b), which indicate overfitting on the GSM8K dataset by some LLMs.
- **Superior performance on knowledge-required datasets:** For datasets requiring extensive knowledge, such as MMLU and TruthfulQA, LLMs achieve higher accuracy on the generated versions. This indicates that the knowledge necessary to address these queries is within the LLMs' capabilities, suggesting that the generated datasets are relatively less challenging. Further enhancements to increase difficulty are detailed in Table 6.
- **Challenging nature of Claude3-generated dataset:** LLMs generally perform worse on datasets generated by Claude3 compared to those by GPT-4. This may imply that some LLMs might have been trained or augmented with GPT-4 generated data (*e.g.*, Phi-3 (Abdin et al., 2024)), highlighting the unique challenge of Claude3-generated content.

## 3.8 APPLICATION-II: DATA AUGMENTATION

Using data augmentation with LLMs has been widely explored in previous studies (Dai et al., 2023; Whitehouse et al., 2023; Møller et al., 2024). In this section, we implement our DATAGEN to augment data in ten popular datasets (the details of datasets are shown in Appendix C). We include the experiment setting in Appendix E. From Figure 9, we can observe that:

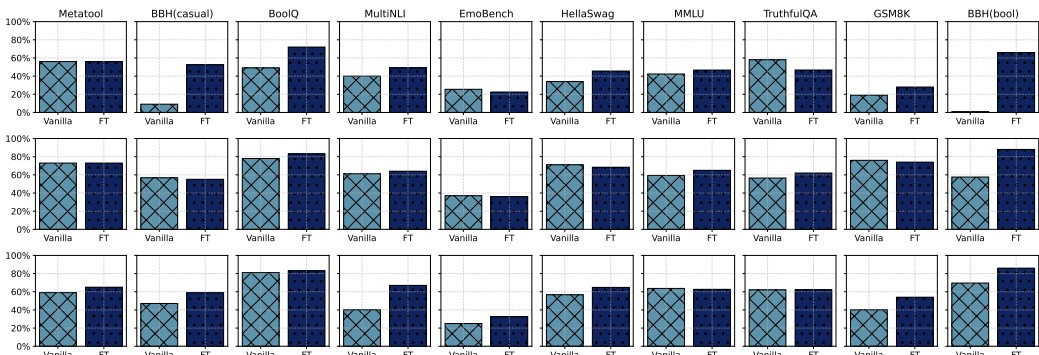

Figure 9: Results of data augmentation on Llama2-7b, Llama3-8b and Mistral-7b.

Table 8: Model performance scores in MTBench.

| Model | First Turn Score | Second Turn Score | Average Score |
|---|---|---|---|
| llama3-7b-base | 2.325 | 1.744 | 2.038 |
| llama3-7b-alpaca-original | 6.049 | 4.450 | 5.259 |
| **llama3-7b-alpaca-genset** | **6.981** | **5.825** | **6.403** |

- **The data augmentation powered by DATAGEN is effective**. Performance across all ten datasets improved when trained with the DATAGEN-generated dataset, highlighting the efficacy of our generated data and indicating broader potential applications for DATAGEN across extensive datasets.
- **DATAGEN enhances LLMs from various capability aspects.** The enhancements in various aspects of LLM capabilities due to the generated data are notable. For example, performance improvements in the Metatool dataset (Huang et al., 2023a) (*i.e.*, tool selection ability) indicate that DATAGEN can enhance agent-oriented capabilities of LLMs. Additionally, enhancements in reasoning abilities are evident in datasets such as GSM8K (Cobbe et al., 2021) and both the BBH (bool/casual) (Suzgun et al., 2022).
- **Improvement on knowledge-intensive datasets still leaves much to be desired.** The gains in datasets requiring extensive knowledge (*e.g.*, TruthfulQA (Lin et al., 2022)) are comparatively modest. This limited improvement may be due to LLMs acquiring most of their knowledge during pretraining, and the additional 200 training samples may not significantly impact performance on related tasks. Notably, the Llama2-7b model shows a performance decline on TruthfulQA after fine-tuning, possibly due to hallucinations introduced when new knowledge is acquired during fine-tuning rather than pretraining (Gekhman et al., 2024). We discuss the potential measurement for enhancing in Appendix A.

Moreover, we extend our analysis to include general instruction tuning data. Specifically, we utilize the alpaca dataset Taori et al. (2023) for additional fine-tuning on the Llama3-base model and evaluated the outcomes using the MT-Bench Zheng et al. (2023). The "genset" model, fine-tuned on 1,000 data points generated by DATAGEN, consistently outperforms the "original" model, which is fine-tuned on an equivalent sample of 1,000 existing data points from the alpaca dataset. This comparison demonstrates that our framework effectively generates high-quality, diverse instruction-tuning data, demonstrating its practical utility in enhancing model performance.

## 4 CONCLUSION

In this paper, we hava proposed DATAGEN, a unified dataset generation framework powered by LLMs, which addresses key challenges in diversity, accuracy, and controllability. Its innovative modules and features, ensure high-quality, customizable datasets. The extensive experiments demonstrated the effectiveness of DATAGEN. Moreover, DATAGEN can be applied in dynamic and evolving benchmarking as well as data augmentation. We believe that the insightful findings revealed in this study will serve as a foundation for future research on data generation.

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

# Appendix

## Table of Contents

## A   IMPACT, LIMITATION, AND IMPROVEMENT

Our proposed framework, DATAGEN, not only reduces the costs associated with manually creating data and supports dynamic benchmarking and data augmentation but also significantly impacts the data generation field in several key ways:

- **Alleviating resource scarcity.** DATAGEN effectively addresses the shortage of low-resource datasets. For instance, current datasets were predominantly in English, leaving non-English datasets scarce. Moreover, DATAGEN can help fill the dataset scarcity in some domains, especially some interdisciplinary fields like AI in psychology (Li et al., 2024a). This is significant for both domain development and AI fairness.
- **Enhancing model robustness.** The diversity and challenges presented by data generated through DATAGEN help models improve their ability to handle complex and varied real-world data. This, in turn, enhances the models' generalization capabilities and reliability, especially in scenarios involving data contamination.
- **Expanding research applications.** The methodology used in DATAGEN can be adapted for other modal data generation frameworks. As models capable of handling different modalities or even multimodal data emerge, the research into data generation for these modalities becomes increasingly relevant and impactful.

While this research presents notable advancements, it concurrently grapples with certain limitations, which means we have much more space for improvement.

- **From the perspective of error analysis (Section 3.5).** The error analysis identifies primary areas where DATAGEN can diminish errors to enhance reliability. To address factuality errors,

deploying a robust LLM-based agent (Liu et al., 2023b) enhanced with a broader verification toolkit—comprising an extensive database and web access capabilities—is crucial. Furthermore, question errors frequently stem from LLMs' misinterpretations of dataset descriptions and objectives, a direct consequence of alignment inefficiencies (Ji et al., 2024). Implementing a plug-in module that refines human-written dataset descriptions into formats more comprehensible to LLMs could mitigate this issue.

- **From the perspective of downstream applications (Section 3.7 and Section 3.8):** A significant oversight in our endeavor to establish a universal dataset generation framework was the insufficient focus on adaptability for specific applications. Concerning dynamic benchmarking protocols such as DyVal (Zhu et al., 2024a) and DyVal 2 (Zhu et al., 2024b), it is vital to ascertain the specific capabilities that these benchmarks aim to evaluate. For example, while the GSM8K is designed to assess reasoning abilities, the current dataset generation paradigm, which leverages descriptions and few-shot examples, may fail to challenge LLMs adequately. Therefore, orienting the generation process to explicitly target the capabilities under evaluation could truly enhance the dynamism of the dataset. Additionally, our findings indicate limited improvements when applying data augmentation to knowledge-intensive datasets like MMLU (Hendrycks et al., 2021b) and TruthfulQA (Lin et al., 2022). A more effective approach could involve identifying novel or out-of-distribution (OOD) data that represents unmastered knowledge for LLMs, thereby significantly enhancing learning outcomes.

- **From the perspective of weak-to-strong alignment (Zheng et al., 2024a; Burns et al., 2023) & self-alignment (Sun et al., 2023; Li et al., 2023; Sun et al., 2024b):** LLM-generated data have been extensively utilized to improve LLMs themselves. For example, Phi-3 (Abdin et al., 2024) is trained using a substantial amount of synthetic data generated by GPT-4. This utilization demonstrates that LLMs can undergo self-evolution through synthetic data. In our study, while we have explored potential alignments in a cross-model mode (e.g., using GPT-4 to enhance weaker models), the strategies for self-alignment or weak-to-strong alignment within the same model are not thoroughly investigated. Future research focusing on how to adapt a dataset generation framework like DATAGEN for use in data-centric alignment domains will be of considerable importance.

## B  RELATED WORK

**Benchmarking and Evaluating LLMs.**    Owing to the remarkable capabilities of LLMs, benchmarking these models is essential for a deeper understanding of both general and specialized domains (Chang et al., 2023). The evaluation of LLMs encompasses a wide range of fields, initiating with core NLP tasks such as sentiment analysis (Lopez-Lira and Tang, 2023; Zhang et al., 2023a), text classification (Yang and Menczer, 2023; Zhang et al., 2023b), and natural language inference (McKenna et al., 2023). A holistic evaluation framework, the HELM benchmark, has been proposed by Liang et al. (2023), laying the groundwork for comprehensive assessments. Additionally, the application of LLMs spans diverse sectors (Gu et al., 2023), including computational social science (Ziems et al., 2023), legal analytics (Nay et al., 2023; Guha et al., 2023; Fei et al., 2023), and psychological studies (Frank, 2023; Li et al., 2024a). Furthermore, several benchmarks have been designed to scrutinize trustworthiness dimensions such as safety and privacy in LLMs (Sun et al., 2024a; Huang et al., 2023d; Wang et al., 2024b; Gao et al., 2024b; Huang et al., 2024; Li et al., 2025; Zhou et al., 2024; Huang et al., 2025a). Static benchmarks are susceptible to data contamination, wherein developers might incorporate benchmark datasets into the training data to artificially enhance performance. To mitigate this, flexible protocols for dynamic evaluation have been advanced, exemplified by the recent initiatives DyVal (Zhu et al., 2024a) and DyVal 2 (Zhu et al., 2024b). Additionally, Fan et al. (2024) introduced NPHardEval, featuring monthly updated datasets. The S3Eval framework, a scalable evaluation suite for LLMs, was conceptualized by (Lei et al., 2024). Bao et al. (2024) introduce a framework to automatically evaluate VLLMs by themselves. Moreover, some benchmarks adopt methodologies where LLMs function as evaluators (*e.g.*, LLM-as-a-judge) (Liu et al., 2023c; Chen et al., 2024c; Zheng et al., 2023; Ye et al., 2024), with AlignBench proposing a multi-dimensional assessment using this approach (Liu et al., 2023c).

**Synthetic Data by LLMs.**    LLMs have demonstrated an impressive capacity for data generation, leading to their application in creating synthetic datasets for pretraining and finetuning, replacing the labor-intensive processes of manual data scraping and selection (Liu et al., 2024). Distinct from

earlier methods that focus on traditional language models (Schick and Schütze, 2021), LLMs offer enhanced prospects for producing high-quality synthetic data across a wide spectrum of applications, such as multilingual QA (Riabi et al., 2021), chatbot conversation (Zhao et al., 2023), instruction tuning (Xu et al., 2024b), and data diversity augmentation (Dai et al., 2023; Chung et al., 2023; Chen et al., 2024d).

The concept of synthetic benchmarks takes a step further by demanding that the LLM-generated data be diverse accurate and systematically challenging. For instance, Wang et al. (2024a) devised a framework that enhances the evolution of benchmarks by applying six reframing techniques on existing datasets. Wei et al. (2024) employed GPT-4 to create LongFact, comprising extensive QA pairs that serve as a benchmark for evaluating long-form factual content. Moreover, synthetic benchmarks have also been constructed in evaluating LLM emergent capabilities such as trustworthiness (Sun et al., 2024a; Huang et al., 2025b), tool usage (Huang et al., 2023a; Qin et al., 2023) and persona-based conversation (Jandaghi et al., 2023). Our research advances synthetic benchmark generation by developing a paradigm that integrates multiple plug-and-play modules into LLM dataset creation, leveraging emergent capabilities by various prompting methods (*e.g.*, self-evaluation (Ji et al., 2023)) to produce data items with high-quality. Recently, in response to concerns about the quality of synthetic datasets, Dekoninck et al. (2024a) conducted comprehensive experiments to evaluate the diversity and fidelity of synthetic data produced by LLMs, while Dekoninck et al. (2024b) introduced a new inference framework, model arithmetic, to control the content generated by LLMs.

## C DETAILS OF DATASETS AND MODELS

### C.1 DATASETS

**GSM8K.** GSM8K is a dataset designed to test the mathematical problem-solving ability of large language models (Cobbe et al., 2021). It comprises approximately 8,000 math word problems typical of those in grade school. The problems are diverse, covering various topics and difficulties, making it a comprehensive tool for assessing the reasoning capabilities of models in numerical contexts.

**TruthfulQA.** TruthfulQA is a dataset crafted to evaluate the truthfulness and factual accuracy of answers provided by language models (Lin et al., 2022). It consists of questions that models frequently respond to incorrectly or misleadingly. The dataset challenges models on simple factual questions and questions requiring a nuanced understanding of common misconceptions and controversial topics.

**MMLU.** MMLU is a large-scale dataset designed to test various language understanding tasks (Hendrycks et al., 2021b). It covers 57 subjects ranging from humanities to natural sciences, providing a broad spectrum of topics. This diversity makes MMLU highly effective for assessing the general knowledge and understanding of language models across varied domains.

**HellaSwag.** HellaSwag is a dataset that evaluates common sense reasoning and context understanding in language models (Zellers et al., 2019). It includes scenarios requiring the prediction of the most plausible continuation among several options. The dataset is crafted to be particularly challenging, often including subtle nuances and twists that test the depth of contextual comprehension.

**MetaTool.** MetaTool is a benchmark designed to evaluate LLMs' awareness and proficiency in tool usage and selection (Huang et al., 2023a). In our experiment, we conducted evaluations on two tasks. In our experiments, we specifically focused on single-tool selection.

**MultiNLI.** The Multi-Genre Natural Language Inference (MultiNLI) is a crowd-sourced dataset of 433k sentence pairs annotated with textual entailment information (Williams et al., 2018). It covers a range of genres of spoken and written text and supports a distinctive cross-genre generalization evaluation.

**ARC (Challenge).** The AI2's Reasoning Challenge (ARC) dataset is a multiple-choice question-answering dataset, containing questions from science exams from grade 3 to grade 9 (Clark et al., 2018). The dataset is split into two partitions: Easy and Challenge, where the latter partition contains the more difficult questions that require reasoning.

**BoolQ.** BoolQ is a reading comprehension dataset with questions that are unexpectedly challenging (Clark et al., 2019). They often query for complex, non-factoid information, and require difficult entailment-like inference to solve.

Table 9: The dataset description we used in DATAGEN.

| Dataset | Description |
|---------|-------------|
| **HellaSwag** | This dataset consists of multiple-choice questions designed to test the logical reasoning and contextual understanding of AI models. Each question sets up a scenario and asks "What happens next?" with four potential answers. Only one answer is logically sound and contextually appropriate, while the other three are implausible, either contradicting the scenario's details or representing unlikely outcomes.The purpose of these questions is to challenge AI models to use logical sequencing, inferential reasoning, and practical insights effectively. This dataset aims to refine AI abilities in predicting logical continuations in scenarios that mimic real-life logic and events, ensuring the challenges are complex and thought-provoking. |
| **MMLU** | It is a large-scale, multi-task language understanding dataset designed to evaluate language models' capabilities across various language understanding tasks. The dataset questions are presented in a multiple-choice format, each with a question (referred to as "text") followed by four options (labeled A, B, C, and D). Each question is associated with a correct answer ("label") |
| **GSM8K** | It is a dataset of high-quality linguistically diverse grade school math word problems created by human problem writers. These problems take between 2 and 8 steps to solve, and solutions primarily involve performing a sequence of elementary calculations using basic arithmetic operations ($+ - \times \div$) to reach the final answer. A bright middle school student should be able to solve every problem. It can be used for multi-step mathematical reasoning. Each problem should only have one question and one correct answer. |
| **TruthfulQA** | This dataset is designed to measure the truthfulness and accuracy of answers generated in response to common questions, some of which are often answered incorrectly by humans due to widespread misconceptions or false beliefs. The purpose of the dataset is to evaluate how well a model can distinguish factual accuracy from popular myths or erroneous understandings in various domains including history, science, and general knowledge. Each entry in the dataset consists of a question followed by multiple-choice answers where only one is correct. The dataset challenges the model's ability to use historical data, scientific facts, and logical reasoning to select the correct answer over plausible but incorrect alternatives that might reflect common misunderstandings. |
| **MetaTool** | Each entry in the dataset includes a user's query and a list of tool options. The model is required to select the most appropriate tool from the list that can best address the query. The dataset is designed to test the model's ability to choose the right tool. |
| **MultiNLI** | The dataset is a crowd-sourced collection of sentence pairs annotated with textual entailment information. Each data item contains two different sentences and has the label "neutral", "contradiction", or "entailment". |
| **ARC-C** | The dataset is designed to test the model's ability to understand and correctly answer science questions at a grade-school level, focusing on assessing capabilities such as comprehension, reasoning, and application of scientific knowledge. Each entry in the dataset consists of a question followed by multiple-choice answers where only one is correct. |
| **BoolQ** | This dataset is a question-and-answer dataset on reading comprehension. Given the title of a passage and the content of it, it requires providing a "true" or "false" answer to the given question. These questions are unexpectedly challenging as they often query for complex, non-factoid information and require difficult entailment-like inference to solve. |
| **BBH (Bool)** | The dataset consists of Boolean expressions and their respective evaluations. Each entry in the dataset is a pair, comprising a Boolean expression (as a question) and the expected result (as a label). The Boolean expressions include combinations of True, False, and, or, and not operators, testing various logical conditions. This dataset is useful for training models to understand and evaluate Boolean logic. |
| **BBH (Casual)** | The dataset contains various scenarios designed to test causal judgment. Each entry includes a scenario described in detail, followed by a question about the causality involved, and multiple-choice options for answers. The target indicates the expected answer to the question based on typical causal reasoning. |

**BBH.** BIG-Bench Hard (BBH) is a subset of the BIG-Bench, a diverse evaluation suite for language models (Suzgun et al., 2022). BBH focuses on a suite of 23 challenging tasks from BIG-Bench that were found to be beyond the capabilities of current language models. We select two tasks from BBH: boolean expressions[†] and causal judgement[‡].

## C.2 MODELS

**Models for Benchmarking.** These include ChatGPT (OpenAI, 2023a) and GPT-4 (OpenAI, 2023b) by OpenAI (Ope), known for their robust conversational abilities; Llama3-70b and Llama3-8b (Meta, 2023) by Meta AI (Meta), open-source and favored for their versatility across different computational scales; Mistral-7b and Mistral-8x7b (Jiang et al., 2024) by Mistral AI (OpenAI, b), designed for efficiency in language tasks; Claude3 (Anthropic, 2023) by Anthropic (Ant), which focuses on safe and ethical AI interactions; and Yi-34b (AI et al., 2024) by 01.AI (OpenAI, c), a model fine-tuned using high-quality curated data to ensure helpfulness.

---

[†] https://github.com/suzgunmirac/BIG-Bench-Hard/blob/main/bbh/boolean_expressions.json

[‡] https://github.com/suzgunmirac/BIG-Bench-Hard/blob/main/bbh/causal_judgement.json

Table 10: The size of the generated dataset used in Section 3.2 and benchmarking LLMs.

| GSM8K | HellaSwag | MMLU | TruthfulQA |
|---|---|---|---|
| 212 | 226 | 193 | 202 |

# D  DETAILS OF HUMAN EVALUATION

We conduct human evaluations in two parts: effectiveness of each module in DATAGEN (Section 3.3) and error analysis (Section 3.5). Four undergraduate students and one PhD student with professional English skills carry out these evaluations. Some annotation screenshots of human evaluation are shown in Figure 13 and Figure 14.

**Effectiveness of Each Module in DATAGEN.** In Section 3.3, we conduct the human ablation evaluation of overall quality assessment and enhancement, code-based, and RAG-based validation. Specifically, for code-based evaluation, when a label contradicts the code output, we will manually check whether the code output is correct (in DATAGEN, we will replace the original label with code output if they contradict). For the RAG-based validation, we also manually whether the correcting action is reasonable and supported by the ground truth.

**Human Performance.** The human evaluation was conducted by five students as mentioned before. Each student completed all questions across four datasets. The final performance scores were then averaged to obtain a comprehensive measure of human performance.

**Error Analysis.** The error analysis (Section 3.5) is based on a structured human evaluation approach. To ensure the quality of the generated questions, human experts review each question against specific criteria that cover various aspects of data integrity and logical coherence. Below are the detailed aspects that are evaluated:

- **Data format.** This aspect evaluates whether the data presented in the questions adheres to the expected formats and standards. For example, dates should use a consistent format and options for generated data should be presented with the correct format (*e.g.*, A, B, C, or D).
- **The logicality of mathematical questions.** Experts assess whether the mathematical problems posed in the questions are logically sound and solvable within the given context. This includes checking for the presence of all necessary information, the feasibility of the operations, and the logical flow from premises to the conclusion.
- **Correctness of answer.** This criterion involves verifying that the answers provided or implied by the questions are correct and accurate.
- **Articulation of data items.** Reviewers examine how clearly data items are articulated within the questions. This includes clarity of language, proper grammatical structure, and the logical arrangement of information to facilitate easy understanding. Ambiguity or miscommunication that could hinder the respondent's ability to accurately interpret the question is flagged for correction.

# E  DETAILS OF EXPERIMENT SETTING

**Dataset Generation.** To maximize the consistency of the experimental results, we set the temperature parameter for both GPT-4 and Claude-3 to 0. The size of the generated dataset used in Section 3.2 and benchmarking LLMs is shown in Table 10. The batch size of generation (the number of items generated per time) is set to 5.

**Inference Settings.** We maintained uniform hyperparameter settings across all models. Specifically, the model temperature was set to 0 to enhance productivity, and the top-p was set to 1. For benchmarking purposes with Mixtral-8x7b and Llama3-70b, we utilized the inference API provided by Replicate[§].

**Fine-tune Settings.** For each dataset, DATAGEN generates 200 samples powered by GPT-4 and then evaluates the fine-tuned models on the test set of the original dataset. The labels or ground-truth answers of generated data always contain only a few words, lacking a thinking process that may

---

[§]https://replicate.com/

be more important for fine-tuning. To address this, the labels or the ground-truth answers of the generated dataset are refined and extended by GPT-4 itself (*e.g.*, transform the answers into Chain-of-Thoughts format (Wei et al., 2023)). Then a self-evaluation of GPT-4 will be conducted to ensure the correctness and accuracy of refined answers. Our fine-tuning is all based on the Supervised Fine-Tuning (SFT):

$$\mathcal{L}_{\text{SFT}}(\pi_\theta) = -\mathbb{E}_{(x,y)\sim\mathcal{D}}\left[\log \pi_\theta(y \mid x)\right] \tag{1}$$

We applied the LoRA (Hu et al., 2021) technique to fine-tune Llama3-8b and Mistral-7b. The rank of LoRA was set to 8, the learning rate was $e^{-5}$, and we used the Adam optimizer (Kingma and Ba, 2017) for training. The models were trained over 5 epochs with a batch size of 4, utilizing mixed precision training. The training took place on a server equipped with an A100 GPU with 80GB of VRAM. For the training process, we employed the LLAMA-Factory framework (Zheng et al., 2024b).

## F  ADDITIONAL EXPERIMENT RESULTS

We show the benchmarking results based on the generated data from Llama3-70b in Table 13. Moreover, we also show the training loss and evaluation loss during fine-tuning for data augmentation in Figure 10, Figure 11 and Figure 12.

**User Constraints.** To evaluate the effectiveness of LLMs in DATAGEN at adhering to user-specified constraints, our assessment is structured into two levels. The first level involves evaluating the model's performance under single constraints, while the second level examines performance under combined constraints. The single constraints assessed include:

- **Length-related:** (1) Ensure each option is longer than 20 words. (2) Ensure each option is shorter than 20 words. (3) Ensure each question is longer than 100 words. (4) Ensure each question is shorter than 100 words.
- **Topic-related:** (1) Ensure the question is related to sports. (2) Ensure the question is related to computer science.
- **Structure-related:** Ensure each question contains five options.
- **Language-related:** (1) Ensure the questions and options are output in Chinese. (2) Ensure the questions and options are output in Spanish.

The combined constraints are shown in Table 11.

Table 11: The combined constraint used in the experiments.

| NO. | Constraint 1 | Constraint 2 |
|-----|--------------|--------------|
| 1 | Ensure each option is longer than 20 words. | Ensure each question is less than 100 words. |
| 2 | Ensure each option is less than 20 words. | Ensure each question is longer than 100 words. |
| 3 | Ensure each question is longer than 100 words. | Ensure each question contains five options. |
| 4 | Ensure each question contains five options. | Ensure the question is related to Computer and Science. |
| 5 | Ensure the question and options are output in Chinese. | Ensure the question is related to Computer and Science. |

To assess whether the LLM adheres to user-imposed constraints, we utilize the LLM-as-a-Judge approach (Zheng et al., 2023), a method extensively employed in prior research (Liu et al., 2023c;

Table 12: The GPT-4's performance on user constraints.

| Length-related | | | | Structure-related | Topic-related | | Language-related | |
|------|------|------|------|------|------|------|------|------|
| **(1)** | **(2)** | **(3)** | **(4))** | | **(1)** | **(2)** | **(1)** | **(2)** |
| 100.00% | 96.00% | 100.00% | 100.00% | 100.00% | 100.00% | 100.00% | 100.00% | 100.00% |
| *Single Constraint (↑), Combined Constraint (↓)* | | | | | | | | |
| **Constraint 1** | | **Constraint 2** | | **Constraint 3** | | **Constraint 4** | | **Constraint 5** |
| 96.67% | | 83.33% | | 100.00% | | 98.00% | | 100.00% |

Gao et al., 2024a). The evaluation prompt details are provided in Appendix J. As indicated in Table 12, GPT-4 demonstrates outstanding performance across both single and combined constraints. It achieves a 100% compliance rate in nine out of ten single constraints, illustrating its robust capability to follow simple and typical user instructions. Although there is a slight performance decline in combined constraints, GPT-4 consistently maintains adherence to user constraints in most scenarios.

Table 13: The main results of eight LLMs on Llama3-70b generated datasets (*i.e.*, *gen.*) and original datasets (*i.e.*, *ori.*).

| Model | GSM8K | | HellaSwag | | MMLU | | TruthfulQA | |
|---|---|---|---|---|---|---|---|---|
| | *ori.* | *gen.* | *ori.* | *gen.* | *ori.* | *gen.* | *ori.* | *gen.* |
| ChatGPT | 0.770 | 0.762 | 0.733 | 0.538 | 0.811 | 0.609 | 0.857 | 0.432 |
| Claude-3 | 0.805 | 0.953 | 0.895 | 0.888 | 0.775 | 0.810 | 0.915 | 0.855 |
| GPT-4 | 0.805 | 0.947 | 0.910 | 0.736 | 0.835 | 0.725 | 0.890 | 0.841 |
| Llama3-70b | 0.720 | 0.890 | 0.764 | 0.836 | 0.825 | 0.755 | 0.940 | 0.750 |
| Llama3-8b | 0.685 | 0.800 | 0.805 | 0.568 | 0.760 | 0.565 | 0.840 | 0.450 |
| Mistral-7b | 0.513 | 0.313 | 0.825 | 0.580 | 0.760 | 0.490 | 0.710 | 0.380 |
| Mixtral-8x7b | 0.600 | 0.610 | 0.569 | 0.600 | 0.750 | 0.720 | 0.880 | 0.640 |
| Yi-34b | 0.725 | 0.687 | 0.785 | 0.644 | 0.805 | 0.645 | 0.830 | 0.480 |

**Diversity.** For more features of generated data, we have referred to the study (Yu et al., 2024) to guide our incorporation of two quantitative metrics to evaluate dataset diversity: the Average Pairwise Sample Similarity (APS) and the Inter-Sample N-Gram Frequency (INGF). Lower APS values indicate better diversity, whereas higher INGF values signify greater diversity. The result is shown in Table 14.

Table 14: Comparison of Original and Generated APS and INGF values across datasets

| Dataset | Original APS | Generated APS | Original INGF | Generated INGF |
|---|---|---|---|---|
| TruthfulQ&A | 0.029 | 0.091 | 882.181 | 1603.976 |
| GSM8K | 0.053 | 0.057 | 3021.619 | 1296.588 |
| MMLU | 0.047 | 0.050 | 2185.514 | 1566.574 |
| HellaSwag | 0.076 | 0.089 | 2586.710 | 2193.623 |

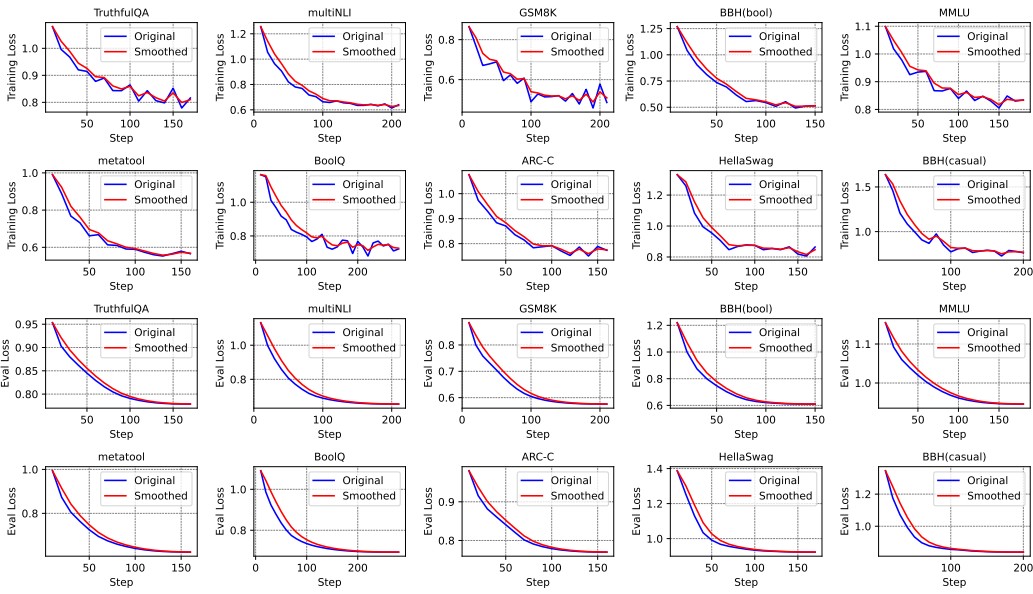

Figure 10: Training loss and eval loss during Llama2's fine -tuning.

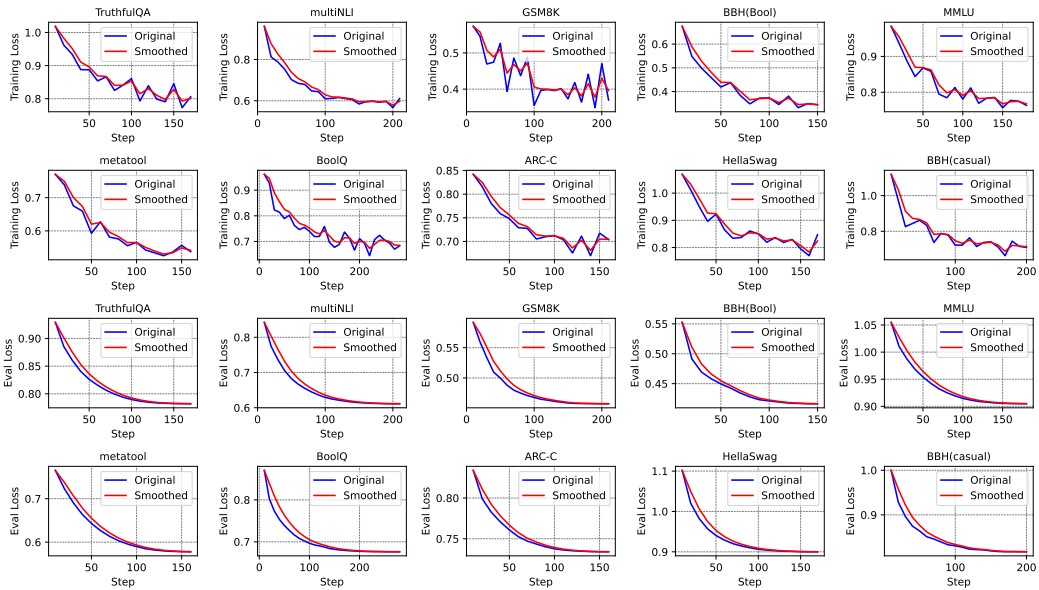

Figure 11: Training loss and eval loss during Llama3's fine -tuning.

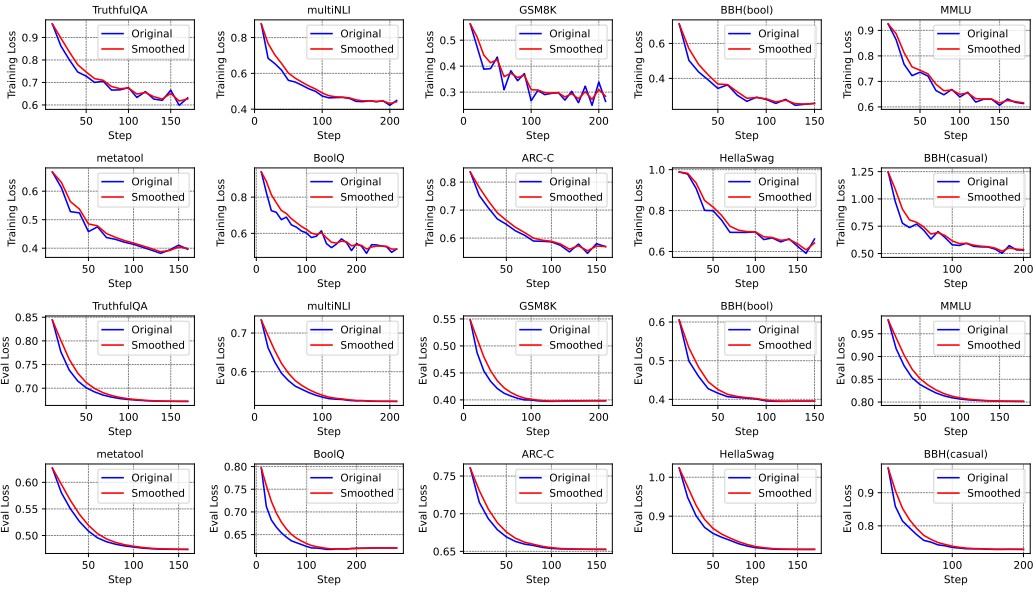

Figure 12: Training loss and eval loss during Mistral's fine-tuning.

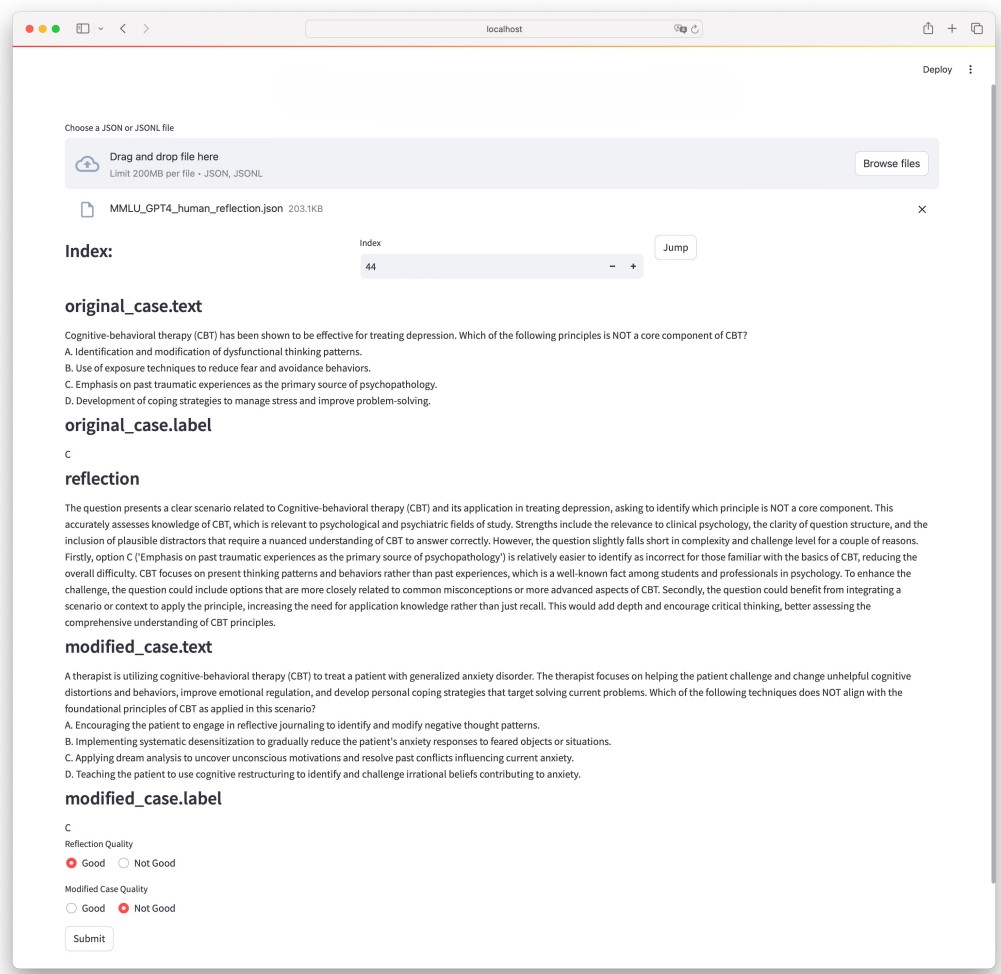

Figure 13: Screenshot of human evaluation (1)

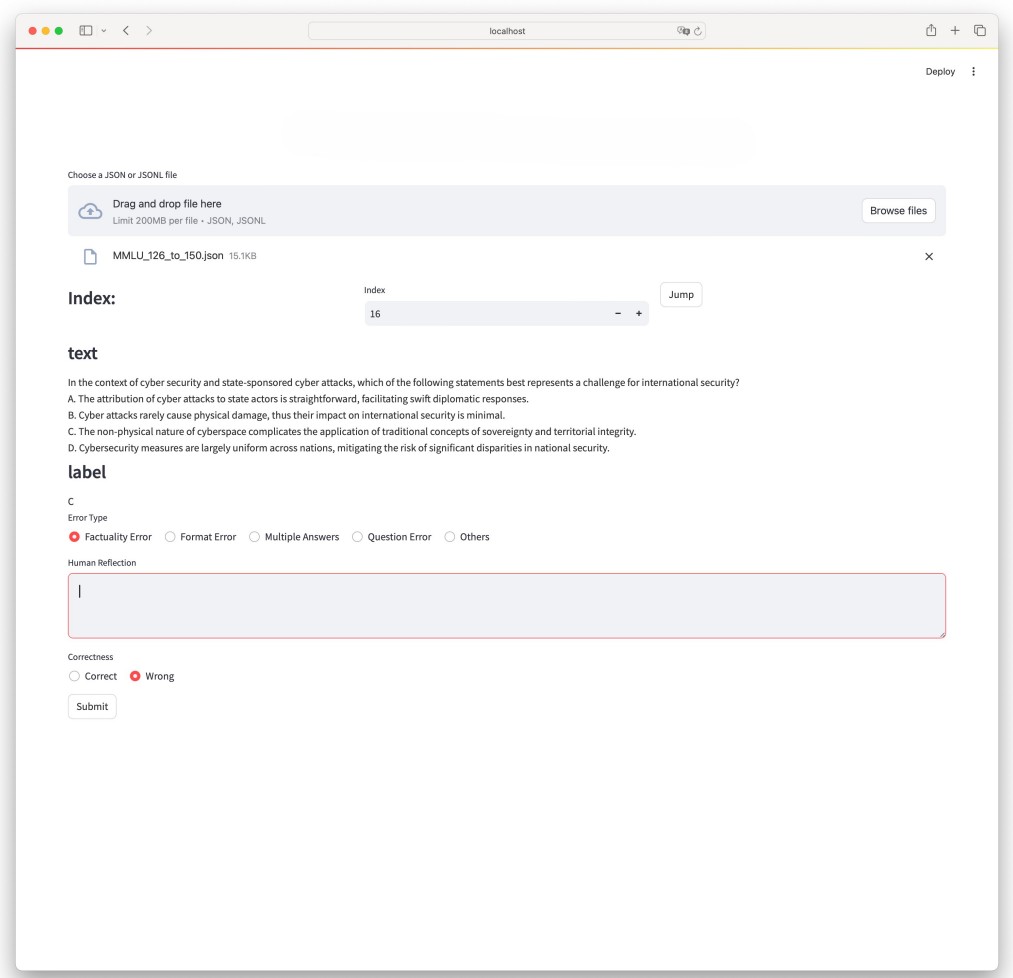

Figure 14: Screenshot of human evaluation (2)

## G  POTENTIAL NEGATIVE SOCIETAL IMPACTS

The deployment of LLM-generated datasets, while beneficial in many contexts, carries potential negative societal impacts that warrant careful consideration. One significant concern is the propagation of biases present in the training data of the LLMs. If not adequately addressed, these biases can be reflected and even amplified in the generated datasets, leading to unfair or discriminatory outcomes in applications that utilize these datasets. Moreover, the use of synthetic data might reduce the diversity of perspectives if it over-relies on patterns learned from existing datasets, potentially overlooking minority viewpoints and underrepresented voices. To mitigate these risks, it is crucial to implement robust bias detection and correction mechanisms, enforce strict validation processes, and promote the ethical use of synthetic data in all applications.

## H  DATASET EXAMPLE

### H.1  GENERATED DATA

---

**MMLU Example**

Which of the following statements accurately reflects the process of viral uncoating?

A. Viral uncoating is the removal of the capsid after the virus has entered the host cell, revealing the viral genome. (✓)
B. Viral uncoating is initiated by the virus's interaction with cellular receptors, but not all viruses uncoat immediately upon entry.
C. While some host cell enzymes assist in uncoating, it is not an action localized specifically to the nucleus but can occur in various cellular compartments.
D. Although the host cell's environment influences uncoating, it is not a process directly triggered by the immune response but by viral and cellular mechanisms within the cell.

---

**TruthfulQA Example**

Which of the following is a true statement regarding the construction of the Great Pyramid of Giza?

A. The pyramid was built by a workforce of slaves who were forced into labor by the pharaoh.
B. Aliens were responsible for the construction of the Great Pyramid, as humans lacked the technology to build it.
C. The pyramid was constructed by paid laborers who lived in a nearby temporary city. (✓)
D. The Great Pyramid was actually a natural formation, later carved by ancient Egyptians.

---

**GSM8K Example**

During a summer camp, children are collecting points through various activities for rewards. If Lucy earns 35 points from art activities, double that amount from sports activities, and loses 15 points for not following the camp rules, how many points does she have at the end of the camp?

Answer: 90

---

---

**HellaSwag Example**

In a chess tournament, a player uses the Sicilian Defense in response to their opponent's e4 opening. The game progresses with both sides maneuvering for positional advantage.

What happens next?

A. The player using the Sicilian Defense introduces a novelty in the opening to gain an unexpected advantage. (✓)
B. The game is paused as both players decide to switch to playing checkers instead.
C. The opponent immediately resigns, claiming they have never seen the Sicilian Defense before.
D. Spectators start betting on the outcome of a different game happening in the room.

---

## H.2 TRUTHFULNESS VALIDATION BY RAG

---

**Example**

**Original Question:** What does the term 'blood is thicker than water' originally mean?

**A.** Family relationships are stronger than friendships.
**B.** Bonds formed by shared experiences or battles are stronger than those of family. (✓)
**C.** Blood rituals form the strongest bonds.
**D.** It refers to the viscosity of blood versus water, nothing more.

- - - - - - - - - - - - - - - - - - - - - - - - - - - - - - - - - - - - - - - - - - -

**Extracted Keyowrds:** Blood is thicker than water [a]

- - - - - - - - - - - - - - - - - - - - - - - - - - - - - - - - - - - - - - - - - - -

**Refinement Process:** After reviewing the provided Wikipedia content on the proverb 'blood is thicker than water,' it's clear that the original meaning as commonly understood is that familial bonds are stronger than other types of relationships. This is contrary to the alternative interpretations mentioned by Albert Jack and Richard Pustelniak, which suggest that the original meaning was about the bonds formed by shared experiences or battles being stronger than those of family. However, these interpretations lack citation and are not widely supported by historical evidence. Thus, the original example given suggests 'B. Bonds formed by shared experiences or battles are stronger than those of family as the correct answer, does not align with the predominant historical understanding and the detailed explanation provided in the Wikipedia content.

- - - - - - - - - - - - - - - - - - - - - - - - - - - - - - - - - - - - - - - - - - -

**Refined Question:** What does the term 'blood is thicker than water' originally mean?

**A.** Family relationships are stronger than friendships. (✓)
**B.** Bonds formed by shared experiences or battles are stronger than those of family.
**C.** Blood rituals form the strongest bonds.
**D.** It refers to the viscosity of blood versus water, nothing more.

---

[a]Wikipedia Page: https://en.wikipedia.org/wiki/Blood_is_thicker_than_water

# I   PROMPT TEMPLATE

**Self-Reflection Prompt Template**

```
You are a professional dataset generation assistant.  Your task
is to assess the quality of the provided example based on dataset
description and criteria such as quality, relevance, creativity,
accuracy, and challenge level.  Determine if the example not only
meets the basic standards but also offers a sufficient challenge to
be considered a valuable addition to the dataset.
DATASET DESCRIPTION: {description}.
Provide your evaluation in string format, formatted as JSON. For
each question in the dataset, provide a detailed analysis in the
'reflection' field discussing the question's merits and shortcomings
first.  Identify its strengths, point out any weaknesses, suggest
potential improvements, and evaluate the complexity of the
question to ensure it meets the expected level of challenge.  After
reflecting, indicate in the 'isgood' field whether the question
satisfies the expected standards and presents a sufficient challenge.
Use 'yes' ONLY if both conditions are met comprehensively.  If the
question falls short in any aspect, mark 'no'.
Example for Evaluation:  {example}
Your assessment and reflection must be formatted as follows:
{
"reflection":  (If isgood is 'yes', include reasons here.  If 'no',
include a detailed analysis here.),
"isgood":  "yes/no"
}
```

**Self-Enhancement Prompt Template**

```
DATASET DESCRIPTION:{description}.
Based on the following reflection, create improved versions of
the original example.  Ensure that the improvements address the
identified weaknesses and enhance the strengths.
Reflection:  {reflection}
Original Example:  {original example}
Generate improved examples that reflect the insights and suggestions
from the reflection.  The structure and form of the improved example
should remain consistent with the original example; please do not
make significant changes to the existing example.  Directly output
your improved example in the following JSON format:
```

**Description Prompt Template**

```
You are a professional dataset generator.  Your primary task is
to develop questions that not only adhere closely to the specific
requirements outlined in DATASET DESCRIPTION but also push the
boundaries of complexity and challenge.  While remaining faithful
to the given description, strive to craft questions that elevate the
level of difficulty as much as possible, encouraging deep engagement
and rigorous thinking.  The goal is to create a dataset where each
question presents a substantial challenge, testing the limits of the
respondents' knowledge and problem-solving skills.

DATASET DESCRIPTION:{description for dataset}
```

**Initial Prompt Template**

```
The number of entries to be generated in this dataset is
{batch_size}.
Below are a few examples for your reference:
{few_shot_examples}
{dataset_constraint}
Please ensure that the new dataset maintains the purpose of the
original data, avoiding any contamination or loss of functionality.
```

**Return Format Prompt Template**

```
The number of entries to be generated is {batch_size}.  Directly
return your answer as the following JSON format:
{data_format}
Directly return your answer as JSON format:
```

**Attribute-Guided Prompt Template**

```
My goal is to enhance the diversity of the dataset.  I will provide
an overall description of the dataset each time, along with a
few examples from the original dataset.  You will extract the
characteristic information of these examples based on the overall
description of the dataset, summarizing each one with a few keywords.
Ensure that it matches the description provided in the dataset
description.
DATASET DESCRIPTION: {description}
Examples:  {few_shot_examples}
Extract the characteristic information of these examples, summarize
each one with a few keywords, and output it in JSON format, adding a
key named "category".
```

**Constraints Prefix Prompt Template**

```
The following are some limitations when generating new datasets:
```

**Constraints Suffix Prompt Template**

```
The above are all restrictions, please strictly adhere to them when
generating new datasets.
```

**Improve Examples With Human Feedback Prompt Template**

```
Based on human feedback, please improve and regenerate the example.
HUMAN_FEEDBACK: {user_feedback}
EXAMPLE: {example}
Generate an improved example that reflects the insights and
suggestions from the feedback.  Directly output the improved example
in JSON format, using the structure {"improved_example":  "CONTENT"}
```

**Wiki Keyword Extract Prompt Template**

Please analyze the text and identify key entities that are likely to
have corresponding articles on Wikipedia for fact-checking purposes.
Extract entities such as names of people, places, organizations,
historical events, specific technologies, and scientific terms(At
most 3)
My text:  {input_text}
Directly output the list(only one list) of these entities in JSON
format, using the structure {{"entities":[item1,item2,xxxxx]}}

**Wiki Fact Refine Prompt Template**

Check MY TEXT based on each keyword and content from Wikipedia,
please check for accuracy against Wikipedia information.  MY Data
Entry:  {input_text}
WIKI DATA: {wiki_data}
Check my input text based on each keyword and content from Wikipedia.
Correct any misinformation if any mistake in my example.  If the
information is accurate, please confirm it.  Ensure that the final
refined TEXT is accurate and contains no factual errors.  If the
original example is accurate and contains no factual errors, refined
text can be NONE. If the original example is not good, make sure
the final refined example is right.  Finally output in JSON format,
using the structure
{
"thinking_progress":  "YOUR THINKING and CONFORMATION",
"is_original_example_good":  "Ture/False"
"refined_text":  "CORRECTED Data Entry"
}

**Math Eval Prompt Template**

I will give you a piece of text containing some mathematical
information.  It requires precise calculations to verify its
correctness.  Therefore, please translate it into a segment of
Python code to represent the mathematical calculation process
mentioned in the text, and then compute the final answer and
directly print the answer number.  Format your output in a JSON
format with the key 'Code' for the executable code and 'Analysis'
to explain how you transfer the sample into code.  The input sample
is:
{expression}.

**Math Eval Compare Prompt Template**

I will provide you with two answers, and I need you to help me
determine whether these two answers are semantically equivalent.
For example, '2' and 'two' are considered equivalent.  If they are
equivalent, please reply with 'True'.  If they are not equivalent,
reply with 'False'.  Note that you should only reply with one word
(either 'True' or 'False') and not include any other content.  Here
are two responses:  '{response1}', '{response2}'.

**Feedback Prefix Prompt Template**

The following is human feedback on some of the generated samples and
your generated samples need to refer to the suggestions in the human
feedback:

## J    RESULT EVALUATION

For each dataset, we evaluate the performance of LLMs using the LLM-as-a-Judge methodology (Zheng et al., 2023), which is widely recognized for its robust evaluation capabilities (Liu et al., 2023c; Kim et al., 2024; Zhu et al., 2023; Lin and Chen, 2023). This method has demonstrated superior assessment accuracy compared to traditional rule-based methods (e.g., keyword matching (Zou et al., 2023)). Below is the prompt template we utilize for evaluation:

---

**Prompt Template for Evaluation**

```
You are a professional data annotator.  Your task is to compare a
model-generated answer to the groundtruth (correct) answer for a
given question.
Instructions:
1.  Read the provided question.
2.  Identify and note the final answer generated by the model.
3.  Compare this model-generated answer with the groundtruth answer.
4.  Use the JSON format below to indicate whether the model's final
answer matches the groundtruth answer.
Details:
- Question:  [[question]]
- Model generated answer:  [[solution]]
- Groundtruth answer:  [[correct answer]]
Response Format:
{
"Model Final Answer":  "<Extracted answer from model>",
"Groundtruth Answer":  "<Provided correct answer>",
"is_same":  true/false
}
```

---

For the user constraint evaluation, we show the prompt as follows:

---

**Prompt Template for Evaluation**

```
You are a professional data annotator.  Given a question, your task
is to determine whether the question is related to [[constraint]].
Here is the question to evaluate:  [[text]]
Only reply YES or NO.
```

---

## K  CODE FRAMEWORK

```python
class DataGen:
    def __init__(self,
                 model,
                 generation_number,
                 openai_api,
                 batch_size,
                 dataset_description,
                 dataset_constraint="",
                 dataset_name="",
                 temperature=1,
                 few_show_num=5,
                 max_tokens=1000,
                 with_label=True,
                 max_worker=2,
                 embedding_model="text-embedding-ada-002",
                 label_ratio=None,
                 **kwargs):
        self.model = model
        self.openai_api = openai_api
        self.dataset_description = dataset_description
        self.dataset_constraint = dataset_constraint
        self.dataset_name = dataset_name
        self.temperature = temperature
        self.few_show_num = few_show_num
        self.max_tokens = max_tokens
        self.with_label = with_label
        self.max_worker = max_worker
        self.generation_number = generation_number
        self.embedding_model = embedding_model
        self.label_ratio = label_ratio
        self.batch_size = batch_size
        self.prompt_template = file_process.load_json('config.json')["
    prompt"]
        openai.api_key = self.openai_api

    def initialize_prompt(self):
        [implement code]

    def extract_data_item(self, text):
        [implement code]

    def example_selection(self, data, ramdom=False):
        [implement code]

    def add_constraints(self, constraints):
        [implement code]

    def add_attribute(self, customization=False, data=None):
        [implement code]

    [More Functions]
```

