# OpenReview forum: "DataGen: Unified Synthetic Dataset Generation via Large Language Models"
_ICLR.cc/2025/Conference — ICLR 2025 Poster_

### Official Review · Reviewer_eQ9J · 2024-10-30

**Soundness:** 3
**Presentation:** 2
**Contribution:** 3
**Rating:** 6
**Confidence:** 4

**Summary:**

This paper proposes a new framework to synthesize high-quality datasets across various types. To ensure dataset quality, the framework integrates an attribute-guided generation module and a group-checking feature to enhance diversity and controllability. It also includes a code-based mathematical assessment and a retrieval-augmented generation module to improve truthfulness and factuality. Experimental results demonstrate the superior quality of the generated datasets in terms of semantics, diversity, and length distribution. By applying the framework to two scenarios, benchmarking LLMs and data augmentation, it validates the effectiveness of this framework.

**Strengths:**

- By integrating different modules, DataGen ensures the quality of generated datasets, considering the diversity, truthfulness, controllability, and so on.
- To assess the quality of generated datasets, the authors design curated experiments and evaluate key factors including length distribution, semantics, diversity, and knowledge richness.
- Experiments on DataGen's effectiveness for data augmentation demonstrate significant benefits across various tasks, particularly in instruction-following scenarios.

**Weaknesses:**

- In Table 1, the authors list a series of current dataset generation frameworks, highlighting that DataGen considers a broader range of factors. However, for downstream applications, especially data augmentation section, none of these methods are compared, which limits the demonstration of their effectiveness.
- As shown in Figure 4(a), while the length distribution of the generated data tends toward a normal distribution, longer-length samples are missing for HellaSwag and MMLU. In Figure 5, although the generated examples align with the original datasets, it is evident that the generated dataset represents only a partial subset of the originals.
- Table 7 shows that, without difficulty enhancements, LLMs perform better on generated benchmarks compared to the originals, which reduces DataGen's effectiveness and practical value.
- The proposed framework is complex, and in the ablation study presented in Table 4, the analysis may be too simplified to fully validate the effectiveness of each module.

**Questions:**

please refer to weaknesses.

---

> ### Author Response · Authors · 2024-11-19
> **Thanks for Your Review**
>
> Thank you so much for your valuable feedback! We truly appreciate your thoughtful suggestions and will address your concerns as follows:
>
> ---
>
> Q: In Table 1, the authors list a series of current dataset generation frameworks, highlighting that DataGen considers a broader range of factors. However, for downstream applications, especially the data augmentation section, none of these methods are compared, which limits the demonstration of their effectiveness.
>
> A: Sorry for the confusion. It is worth emphasizing that in Table 1, we conducted a broad and comprehensive comparison of current dataset generation frameworks across eight dimensions. DataGen stands out because its capabilities span all eight dimensions:
>
> 1. It not only supports data augmentation but also dynamic evaluation, introducing new knowledge to the generated datasets without requiring manual intervention.
> 2. We designed several modules to ensure high-quality data generation, focusing on truthfulness and diversity.
>
> **Compared to other dataset generation frameworks, the key advantage of DataGen lies in its strong generalizability and flexibility. It is not solely limited to data augmentation, as we aim to strike an optimal balance between generalizability and effectiveness. For example, while frameworks like MetaMath and Dyval can be used for data augmentation, they are primarily focused on reasoning datasets rather than all types of datasets.**
>
> In summary, we plan to explore further optimization of this balance in future work. Thank you very much for your suggestion!
>
> ---
>
> Q: As shown in Figure 4(a), while the length distribution of the generated data tends toward a normal distribution, longer-length samples are missing for HellaSwag and MMLU. In Figure 5, although the generated examples align with the original datasets, it is evident that the generated dataset represents only a partial subset of the originals.
>
> A: Thank you very much for pointing this out! To address your concern, we injected specific instructions into the dataset constraints to encourage LLMs to generate longer samples. The results of this approach are as follows:
>
> | Dataset | Text Length | Word Count |
> | --- | --- | --- |
> | MMLU (Original) | 559.83 | 92.38 |
> | MMLU (Long) | 1154.31 | 167.51 |
> | HellaSwag (Original) | 797.59 | 146.55 |
> | HellaSwag (Long) | 1003.16 | 145.88 |
>
> It can be seen that by introducing constraints, LLM effectively generate longer text, which means the controlability of DataGen. This operation can be achieved by modifying the configuration file in our toolkit (e.g., datagen/examples/generation_config.yaml).
>
> ---
>
> Q: Table 7 shows that, without difficulty enhancements, LLMs perform better on generated benchmarks compared to the originals, which reduces DataGen's effectiveness and practical value.
>
> A: Thank you very much for your suggestion. By default, it is challenging to make LLMs generate more difficult data, which is why we introduced difficulty enhancements. It is important to emphasize that difficulty enhancements are an integral part of the DataGen framework, not an isolated feature. Users can configure these enhancements based on their specific needs (as README in toolkit shows). It may not be entirely fair to evaluate benchmark difficulty solely by the presence or absence of difficulty enhancements. This might have been a writing issue in our paper, and we have now clarified this point in the revised version. Thank you again for pointing this out!
>
> ---

---

> ### Author Response · Authors · 2024-11-19
> **Thanks for Your Review**
>
> Q: The proposed framework is complex, and in the ablation study presented in Table 4, the analysis may be too simplified to fully validate the effectiveness of each module.
>
> A: Thank you very much for your suggestion! Due to the structure of our paper, which focuses on the procedule by order of the framework's modules, our ablation studies are not limited to Table 4. For instance:
>
> - In Figure 3, we present the ablation results of self-reflection module.
> - In Figure 8, we analyze six ablation scenarios from a cost perspective.
>
> We appreciate your feedback and have updated the caption of Figure 3 and the title of Section 3.6 to reduce any confusion.
>
> Additionally, to address your concern thoroughly, we conducted additional experiments based on your suggestion, including code-based mathematical validation, RAG module evaluation, diversity measurement and iteration number of self-reflection. The results are as follows:
>
> Code-Based Mathematical Validation:
>
> | Aspect | Llama3-70b | Claude-3 | Llama3-405b |
> | --- | --- | --- | --- |
> | Code | 100% | 97.96% | 90.53% |
> | Original | 39.88% | 20.43% | 33.68% |
>
> RAG Module Evaluation:
>
> | RAG | GPT-4 | Llama3-70b | Claude-3 | Llama3-405b |
> | --- | --- | --- | --- | --- |
> | Improvement (%) | 4.20% | 22.35% | 24.62% | 4.05% |
>
> Diversity Measurement:
>
> | Model | Type | GSM8K | TruthfulQA | HellaSwag | MMLU |
> | --- | --- | --- | --- | --- | --- |
> | Claude-3 | Generated | 0.676 | 0.758 | 0.685 | 0.748 |
> |  | Original | 0.682 | 0.745 | 0.743 | 0.746 |
> |  | Δ | 0.88% | 1.74% | 7.81% | 0.27% |
> | Llama3-70b | Generated | 0.621 | 0.732 | 0.705 | 0.741 |
> |  | Original | 0.682 | 0.745 | 0.743 | 0.746 |
> |  | Δ | 8.94% | 1.74% | 5.11% | 0.67% |
> | Llama3-405b | Generated | 0.643 | 0.633 | 0.666 | 0.647 |
> |  | Original | 0.682 | 0.745 | 0.743 | 0.746 |
> |  | Δ | 5.72% | 15.03% | 10.36% | 13.27% |
>
>
> Iteration Number of Self-Reflection (%):
>
> | Dataset | 1 | 2 | 3 | 4 | 5 | 5+ |
> | --- | --- | --- | --- | --- | --- | --- |
> | HellaSwag (Llama3-70b) | 89.38 | 10.26 | 0.37 | 0.00 | 0.00 | 0.00 |
> | HellaSwag (Claude-3) | 74.54 | 15.74 | 6.48 | 2.31 | 0.93 | 0.00 |
> | TruthfulQA (Llama3-70b) | 88.76 | 6.74 | 2.81 | 0.56 | 0.00 | 1.12 |
> | TruthfulQA (Claude-3) | 81.07 | 16.02 | 1.46 | 0.97 | 0.49 | 0.00 |
> | GSM8K (Claude-3) | 96.94 | 3.06 | 0.00 | 0.00 | 0.00 | 0.00 |
> | GSM8K (Llama3-70b) | 68.45 | 24.40 | 6.55 | 0.60 | 0.00 | 0.00 |
> | MMLU (Claude-3) | 90.29 | 8.86 | 0.86 | 0.00 | 0.00 | 0.00 |
> | MMLU (Llama3-70b) | 71.57 | 15.74 | 7.36 | 4.31 | 0.76 | 0.25 |
>
> From the above results, we observe:
>
> 1. **The DataGen framework exhibits strong generalizability.** Regardless of whether the model is open-weight or proprietary, DataGen performs effectively across all modules.
> 2. **DataGen significantly enhances the truthfulness of data generation for many models.** For example, DataGen reduces mathematical generation errors for Llama3-70b, Claude-3, and Llama3-405b by over 50%. Additionally, the RAG module effectively mitigates factual errors for Claude-3 and Llama3-70b, with a correction rate exceeding 20%.
>
> ---
>
> Thank you again for your valuable suggestions. We hope our responses address your concerns thoroughly and effectively!

---

> > ### Comment · Reviewer_eQ9J · 2024-11-27
> >
> > Thanks for the authors' detailed responses, which address my concerns greatly.
> >
> > I will increase my overall rating to 6.

---

> > > ### Author Response · Authors · 2024-11-27
> > > **Thanks For Your Review**
> > >
> > > Thank you so much for acknowledging our work! Your encouraging feedback has given us renewed motivation and strengthened our confidence in our efforts. On behalf of all the authors, I would like to express our heartfelt gratitude to you.

---

> ### Author Response · Authors · 2024-11-27
> **Thanks For Your Review**
>
> Dear Reviewer,
>
> Thank you very much for taking the time to review our paper. If you have a moment, could you kindly confirm whether our responses have addressed your concerns? Thank you so much!

---

### Official Review · Reviewer_S9ep · 2024-11-02

**Soundness:** 3
**Presentation:** 3
**Contribution:** 3
**Rating:** 6
**Confidence:** 4

**Summary:**

The paper introduces DateGen that uses LLM to generate synthetic dataset. DateGen overcomes limitations in generalization, controllability, diversity, and truthfulness, DATAGEN supports a variety of dataset formats and includes mechanisms like attribute-guided generation and group-checking to enhance diversity. It also employs mathematical code-based assessment and Retrieval-Augmented Generation (RAG) for accuracy and truthfulness. Experimentation confirms superior data quality, with applications in benchmarking LLMs and data augmentation, leading to improved model performance in domains like reasoning and agent capabilities

**Strengths:**

1.  DataGen introduces novel elements like attribute-guided generation and the RAG-based validation, which distinguish it from existing synthetic dataset generation frameworks.
2. The modular design allows for customization and adaptability across diverse datasets.
3. The experiments with improved reasoning and agent-oriented tasks performance shows potential in this data generation framework.

**Weaknesses:**

1. RAG-based validation is very high in cost (raising cost from 0.038 to 0.19, almost 5x increase). However, it is unclear how it affects the final data generation quality (like the results in Table 7). In other words, it would be nicer to ablate the modules in terms of metrics in Table 7, instead of the current reports in Table 4.
2. I am not convinced that the performance decline on GSM8K in your experiments can be concluded to that many LLMs may be overstated and overfit on the GSM8K dataset, would you please elaborate more on this?
3. Have you tried any experiments with open-sourced LLMs such as LLaMA 405B being the generating LLM? Would it be as beneficial as the GPT-4 or Claude?

**Questions:**

My questions are included in the weakness section.

---

> ### Author Response · Authors · 2024-11-19
> **Thanks for Your Review**
>
> Thank you very much for your valuable feedback! We will address your concerns one by one:
>
> ---
>
> Q: RAG-based validation is very high in cost (raising cost from 0.038 to 0.19, almost a 5x increase). However, it is unclear how it affects the final data generation quality (like the results in Table 7). In other words, it would be better to ablate the modules in terms of metrics in Table 7, instead of the current reports in Table.
>
> A: This is an good suggestion! Regarding your concern about the high cost of RAG-based validation, there are indeed several alternative solutions:
>
> 1. Using a cheaper model: We can substitute the current GPT-4o with a more cost-effective model like GPT-4o-mini, which scores 82% on MMLU and currently outperforms GPT-4 on chat preferences in the [LMSYS leaderboard](https://arena.lmsys.org/). This substitution could **reduce RAG costs by 16x, making the cost decrease from 0.19 to 0.048**, which is entirely acceptable. We have also included options for different models in the RAG module of our toolkit.
> 2. Prompt compression techniques: There are many established and effective techniques for prompt/document compression [1][2][3], which we plan to integrate into our toolkit to further reduce RAG costs.
>
> Regarding your concern about the unclear impact of RAG on the results in Table 7, we want to emphasize that Table 7 presents benchmark results for each model, where RAG's effects may not be directly reflected. Nevertheless, we conducted ablation studies on the RAG module on four models and found that adding RAG reduced factual errors significantly, which is critical for improving dataset quality. Here are the results:
>
> | RAG | GPT-4 | Llama3-70b | Claude-3 | Llama3.1-405b |
> | --- | --- | --- | --- | --- |
> | Improvement (%) | 4.20% | 22.35% | 24.62% | 4.05% |
>
> ---
>
> Q: I am not convinced that the performance decline on GSM8K in your experiments can be concluded to the claim that many LLMs may be overstated and overfit on the GSM8K dataset. Would you please elaborate more on this?
>
> A: Sorry for the confusion! We believe that LLMs exhibit overfitting on GSM8K, which, in other words, implies a risk of data leakage. This has been highlighted in many recent studies [4], and similar research has reached conclusions consistent with ours [5, 6]. We will add more citations to this finding in PDF.
>
> ---

---

> ### Author Response · Authors · 2024-11-19
> **Thanks for Your Review**
>
> Q: Have you tried any experiments with open-sourced LLMs such as LLaMA 405B being the generating LLM? Would it be as beneficial as GPT-4 or Claude?
>
> A: We sincerely apologize for any confusion caused. Due to previous computational constraints, we were unable to conduct experiments with Llama3-405B. However, we did evaluate open-sourced models, such as Llama3-70B, on DataGen, and the benchmark results are presented in the appendix as follows:
>
> | Model | GSM8K (ori.) | GSM8K (gen.) | HellaSwag (ori.) | HellaSwag (gen.) | MMLU (ori.) | MMLU (gen.) | TruthfulQA (ori.) | TruthfulQA (gen.) |
> | --- | --- | --- | --- | --- | --- | --- | --- | --- |
> | ChatGPT | 0.770 | 0.762 | 0.733 | 0.538 | 0.811 | 0.609 | 0.857 | 0.432 |
> | Claude-3 | 0.805 | 0.953 | 0.895 | 0.888 | 0.775 | 0.810 | 0.915 | 0.855 |
> | GPT-4 | 0.805 | 0.947 | 0.910 | 0.736 | 0.835 | 0.725 | 0.890 | 0.841 |
> | Llama3-70b | 0.720 | 0.890 | 0.764 | 0.836 | 0.825 | 0.755 | 0.940 | 0.750 |
> | Llama3-8b | 0.685 | 0.800 | 0.805 | 0.568 | 0.760 | 0.565 | 0.840 | 0.450 |
> | Mistral-7b | 0.513 | 0.313 | 0.825 | 0.580 | 0.760 | 0.490 | 0.710 | 0.380 |
> | Mixtral-8x7b | 0.600 | 0.610 | 0.569 | 0.600 | 0.750 | 0.720 | 0.880 | 0.640 |
> | Yi-34b | 0.725 | 0.687 | 0.785 | 0.644 | 0.805 | 0.645 | 0.830 | 0.480 |
>
> We highly value your suggestion and subsequently acquired additional computational resources to conduct experiments with Llama3-405B. The results are as follows:
>
> | Model | GSM8K (ori.) | GSM8K (gen.) | HellaSwag (ori.) | HellaSwag (gen.) | MMLU (ori.) | MMLU (gen.) | TruthfulQA (ori.) | TruthfulQA (gen.) |
> | --- | --- | --- | --- | --- | --- | --- | --- | --- |
> | ChatGPT | 0.770 | 0.497 | 0.733 | 0.757 | 0.811 | 0.546 | 0.857 | 0.294 |
> | GPT-4 | 0.805 | 0.546 | 0.910 | 0.637 | 0.835 | 0.613 | 0.890 | 0.394 |
> | Llama3-70b | 0.720 | 0.562 | 0.764 | 0.757 | 0.825 | 0.851 | 0.940 | 0.894 |
> | Llama3-8b | 0.685 | 0.476 | 0.805 | 0.768 | 0.760 | 0.825 | 0.840 | 0.894 |
> | Mixtral-8x7b | 0.600 | 0.364 | 0.569 | 0.681 | 0.750 | 0.479 | 0.880 | 0.333 |
> | Mistral-7b | 0.513 | 0.316 | 0.825 | 0.724 | 0.760 | 0.449 | 0.710 | 0.361 |
> | Yi-34b | 0.725 | 0.645 | 0.785 | 0.795 | 0.805 | 0.794 | 0.830 | 0.722 |
>
> As shown, open-sourced models effectively achieve dynamic evaluation under the DataGen framework. Many models show significant performance drops on generated data, revealing potential overfitting to test datasets [4]. This demonstrates the effectiveness of DataGen in dynamic benchmarking. If the paper is accepted, we will include these results in the camera-ready version.
>
> ---
>
> Thank you again for your valuable feedback, especially your insightful analysis regarding GSM8K overfitting. We sincerely hope our responses address your concerns!
>
> [1] Huiqiang Jiang, Qianhui Wu, Xufang Luo, Dongsheng Li, Chin-Yew Lin, Yuqing Yang, and Lili Qiu. 2024. [LongLLMLingua: Accelerating and Enhancing LLMs in Long Context Scenarios via Prompt Compression](https://aclanthology.org/2024.acl-long.91). In *Proceedings of the 62nd Annual Meeting of the Association for Computational Linguistics (Volume 1: Long Papers)*, pages 1658–1677, Bangkok, Thailand. Association for Computational Linguistics.
>
> [2] Zhuoshi Pan, Qianhui Wu, Huiqiang Jiang, Menglin Xia, Xufang Luo, Jue Zhang, Qingwei Lin, Victor Rühle, Yuqing Yang, Chin-Yew Lin, H. Vicky Zhao, Lili Qiu, and Dongmei Zhang. 2024. [LLMLingua-2: Data Distillation for Efficient and Faithful Task-Agnostic Prompt Compression](https://aclanthology.org/2024.findings-acl.57). In *Findings of the Association for Computational Linguistics: ACL 2024*, pages 963–981, Bangkok, Thailand. Association for Computational Linguistics.
>
> [3] Huiqiang Jiang, Qianhui Wu, Chin-Yew Lin, Yuqing Yang, and Lili Qiu. 2023. [LLMLingua: Compressing Prompts for Accelerated Inference of Large Language Models](https://aclanthology.org/2023.emnlp-main.825). In *Proceedings of the 2023 Conference on Empirical Methods in Natural Language Processing*, pages 13358–13376, Singapore. Association for Computational Linguistics.
>
> [4] Mirzadeh, Iman, et al. "Gsm-symbolic: Understanding the limitations of mathematical reasoning in large language models." *arXiv preprint arXiv:2410.05229* (2024).
>
> [5] Zhu, Kaijie, et al. "Dyval: Dynamic evaluation of large language models for reasoning tasks." *The Twelfth International Conference on Learning Representations*. 2023.
>
> [6] Zhang, Hugh, et al. "A careful examination of large language model performance on grade school arithmetic." arXiv preprint arXiv:2405.00332 (2024).

---

> > ### Comment · Reviewer_S9ep · 2024-11-24
> >
> > Thank you for your dedicated response. It clears some of my concerns. And I would like to keep my score (6) unchanged.

---

### Official Review · Reviewer_1kEc · 2024-11-03

**Soundness:** 3
**Presentation:** 3
**Contribution:** 3
**Rating:** 6
**Confidence:** 3

**Summary:**

This paper introduces DataGen, a comprehensive framework for generating high-quality (diverse, accurate, and controllable) datasets using large language models (LLMs). DataGen accepts diverse dataset and constraints as input, a comprehensive set of generation hints to reduce computational cost, augment diversity with hyperparameter setting / attribute guided generation, and increase evaluation quality using various reasoning techniques (self-refine) / strong code-based verifier / RAG. Evaluation shows DataGen is able to generalize to diverse set of domains and tasks, models.

**Strengths:**

**Novelty and Significance**. The paper presents a novel technique and artifact for the field of synthetic data generation. DataGen is generalizable to other domains and tasks, though with additional overhead. Compared to other related work, DataGen is able to cover a wide range of features in real settings. The artifact is available and runnable.

**Writing**. The writing is clear and well-organized, with clear visual / tables to summarizes the comparison, methodology, evaluation, and ablation studies. The motivation of the paper is very clear, the problem is well-defined, key contributions are listed and aligned with the structure of the paper. The visual elements in the paper are very helpful to understand the paper.

**Methodology**. The proposed framework is well-motivated, and the framework design is simple and easily generalizable to different domains and setups.

**Evaluation**. The evaluation is very comprehensive. It covers a wide range of tasks, models (open and closed source), and domains.

**Weaknesses:**

**Data formatting.** In section 3.5 (error analysis), the paper mention sometimes the dataset strggles to follow instruction / format the data correctly. Using constrained decoding and similar techniques, this is a very much solved problem, but produce result that the LLM itself may not follow (hence potentially dropping quality of response). I recommend checking out related works in this field (e.g. Guidance[1], AICI[2], LMQL[3], etc.) to improve the data formatting issue. Further more, LLM engiens such as vLLM[4], SGLang[5] and other proprietary engines (Anthropic, Gemini) have provide structured output generation to support constrained decoding at the time of data generation.

[1] https://github.com/guidance-ai/guidance
[2] https://github.com/microsoft/aici
[3] https://lmql.ai/
[4] https://vllm.ai/
[5] https://github.com/sgl-project/sglang


**Section 3.3 Effectiveness of Modules in DataGen**.
- Can you explain more on remote-clique score? Why is it a good metric, how exactly is it calculated on the generated dataset (using embeddings, or other representation of the dataset)?
- The delta of remote-clique score of HellaSwag is significantly higher than other datasets. Why is that?

**Questions:**

**Section 3.8 Benchmarking LLM**. The paper mentioned "challenging nature of Claude-3 generated dataset". Do different LLM uses the same / different prompts?

**Section 3.3 Effectiveness of Modules in DataGen**.
- Can you explain more on remote-clique score? Why is it a good metric, how exactly is it calculated on the generated dataset (using embeddings, or other representation of the dataset)?
- The delta of remote-clique score of HellaSwag is significantly higher than other datasets. Why is that?

---

> ### Author Response · Authors · 2024-11-19
> **Thanks for Your Review**
>
> Thank you for your thoughtful feedback and valuable suggestions! We hope the following responses can address your concern:
>
> ---
>
> **Q: Data formatting.**
>
> **A:** We sincerely apologize for the misunderstanding. The "data format error" mentioned does not only refer to non-compliance with JSON but also to issues such as improper output formatting or missing information, such as generating only the question without the answer. Regarding the resolution of JSON formatting issues, we mentioned in line 407 of our paper that we employed "an integrated framework like LangChain." Thank you for your suggestion! We are in the process of updating our toolkit, and once the paper is accepted, we will release a new version of the toolkit. This update will support three formatting frameworks: `Guidance`, `LangChain`, and OpenAI's native JSON interface.
>
> ---
>
> **Q: Effectiveness of Modules in DataGen.**
>
> **A:** The Remote-Clique Score (RCS) is a robust metric because it directly measures semantic cohesion among texts using embeddings, focusing on closely related pairs that exceed a similarity threshold. Here's how it is calculated:
>
> 1. **Get Embeddings**: Each text is converted into a vector $E_i$ using a model (in our case, OpenAI's `text-embedding-ada-002`).
> 2. **Cosine Similarity**:
>
>     $$S(i, j) = \frac{E_i \cdot E_j}{\|E_i\| \|E_j\|}$$
>
>     Similarity between text pairs is computed.
>
> 3. **Form Cliques**:
>     - A threshold $\theta$ is defined.
>     - Texts are considered part of the same clique if $S(i, j) > \theta$.
> 4. **Compute RCS**:
>
>     $$\text{RCS} = \frac{\sum_{i < j, S(i, j) > \theta} S(i, j)}{N}$$
>
>     Here, $N$ is the number of text pairs above the threshold.
>
>
> We referenced prior work [1], which also used this metric to evaluate text diversity. Additionally, we incorporated other diversity metrics to ensure comprehensive evaluation, such as APS and INGF. The results, presented in Appendix Table 14, are as follows:
>
> | Dataset | Original APS | Generated APS | Original INGF | Generated INGF |
> | --- | --- | --- | --- | --- |
> | TruthfulQ&A | 0.029 | 0.091 | 882.181 | 1603.976 |
> | GSM8K | 0.053 | 0.057 | 3021.619 | 1296.588 |
> | MMLU | 0.047 | 0.050 | 2185.514 | 1566.574 |
> | HellaSwag | 0.076 | 0.089 | 2586.710 | 2193.623 |
>
> Regarding why HellaSwag shows a larger difference in the remote-clique score compared to other datasets: the difference here does not indicate improvement but rather the diversity difference between the original and generated datasets. The 8% difference shows that HellaSwag has the largest diversity gap, though this is still objectively less than 10%, which is acceptable. This larger gap might be attributed to the dataset's longer average text length. During generation, the model may have sacrificed some diversity to maintain length distribution.
>
> ---
>
> **Q: Benchmarking LLM.**
>
> **A:** We apologize for any confusion caused. In our experiments, we used the same prompt for all generators to ensure a fair comparison (this clarification has been revised in the PDF to reduce misunderstandings). One reason Claude-generated questions might be perceived as harder is that many models are fine-tuned or pre-trained on synthetic data generated by GPT-4, making them more familiar with GPT-style content, which could create difficulty discrepancies. If user would like to experiment with different prompts, they can modify the configurations in our toolkit at `datagen/utils/prompt.py`.
>
> ---
>
> Thank you once again for your valuable feedback. We hope our responses address your concerns effectively!
>
> [1] Li, Zhuoyan, et al. "Synthetic Data Generation with Large Language Models for Text Classification: Potential and Limitations." *The 2023 Conference on Empirical Methods in Natural Language Processing*.

---

> > ### Comment · Reviewer_1kEc · 2024-11-26
> > **Thank you**
> >
> > This addresses my concern and comments. Thank you for your detailed explanation!

---

### Official Review · Reviewer_WuNv · 2024-11-04

**Soundness:** 3
**Presentation:** 3
**Contribution:** 3
**Rating:** 6
**Confidence:** 3

**Summary:**

The authors present a framework for generating synthetic datasets that focus on generalization, controllability, diversity, and truthfulness by guiding the generation with attributes, checking diversity within a clique, performing code-based verification for reasoning tasks, and performing RAG to verify facts. The authors also show what types of synthetic benchmarks LLMs excel at and fail at.

**Strengths:**

- There's comprehensive work into each of the target attributes (generalization, controllability, diversity, and truthfulness)
- The methodology is highly detailed, including comprehensive ablations, evaluations, and cost details.
- The details about what synthetic generations other LLMs perform well and poorly on are helpful for further work into synthetic benchmarks.

**Weaknesses:**

- There could be more side-by-sides of questions from the original dataset and each generated dataset.

**Questions:**

- What is the performance of each module given different generator models?

---

> ### Author Response · Authors · 2024-11-19
> **Thanks for Your Reivew**
>
> Thank you very much for your suggestions! We will address your concerns one by one:
>
> ---
>
> **Q: There could be more side-by-side of questions from the original dataset and each generated dataset.**
>
> **A:** We are sorry for the confusion. We did not include side-by-side generation in our experiments mainly for the following reasons:
>
> 1. **Token Consumption.** The efficiency of generating side-by-sides is relatively low, increasing our framework's token consumption during the generation process.
> 2. **The Same Goal as the Current Pipeline.** The generation of side-by-side shares the same goal with our current generation process. Both approaches focus on generating questions that deviate from the original dataset's goals and task objectives while introducing new knowledge.
> 3. **Still Supported By DataGen.** Despite these considerations, our framework still supports side-by-side. By modifying the configuration file in our toolkit (e.g., `datagen/examples/generation_config.yaml`), you can specify the requirement for side-by-sides in the `constraint` section (`generation_hint/dataset_constraint`). The LLM can then generate side-by-side data as per the requirements. We have demonstrated the LLM's performance on instruction-following under given constraints (for single and multiple constraints) in the appendix:
>
> |  | Length-related |  |  |  | Structure-related |  | Topic-related |  | Language-related |  |
> | --- | --- | --- | --- | --- | --- | --- | --- | --- | --- | --- |
> |  | (1) | (2) | (3) | (4) | (1) |  | (1) | (2) | (1) | (2) |
> | Percentage | 100.00% | 96.00% | 100.00% | 100.00% | 100.00% |  | 100.00% | 100.00% | 100.00% | 100.00% |
>
> | Constraint 1 | Constraint 2 | Constraint 3 | Constraint 4 | Constraint 5 |
> | --- | --- | --- | --- | --- |
> | 96.67% | 83.33% | 100.00% | 98.00% | 100.00% |
>
> From these results, it is evident that under the DataGen framework, LLMs perform exceptionally well in instruction-following, which shows potential capability on side-by-side generation.
>
> ---
>
> **Q: What is the performance of each module given different generator models?**
>
> **A:** Thanks for your suggestion! Based on your feedback, we supplemented relevant experiments. Specifically, we evaluated code-based mathematical validation, diversity measurement, RAG module, and the number of iterations in self-reflection. The results are as follows:
>
> **Code-Based Mathematical Evaluation:**
>
> | Aspect | Llama3-70b | Claude-3 | Llama3.1-405b |
> | --- | --- | --- | --- |
> | Code | 100% | 97.96% | 90.53% |
> | Original | 39.88% | 20.43% | 33.68% |
>
> **Diversity Measurement:**
>
> | Model | Type | GSM8K | TruthfulQA | HellaSwag | MMLU |
> | --- | --- | --- | --- | --- | --- |
> | Claude-3 | Generated | 0.676 | 0.758 | 0.685 | 0.748 |
> |  | Original | 0.682 | 0.745 | 0.743 | 0.746 |
> |  | Δ | 0.88% | 1.74% | 7.81% | 0.27% |
> | Llama3-70b | Generated | 0.621 | 0.732 | 0.705 | 0.741 |
> |  | Original | 0.682 | 0.745 | 0.743 | 0.746 |
> |  | Δ | 8.94% | 1.74% | 5.11% | 0.67% |
> | Llama3.1-405b | Generated | 0.643 | 0.633 | 0.666 | 0.647 |
> |  | Original | 0.682 | 0.745 | 0.743 | 0.746 |
> |  | Δ | 5.72% | 15.03% | 10.36% | 13.27% |
>
> **RAG Module:**
>
> | RAG | GPT-4 | Llama3-70b | Claude-3 | Llama3.1-405b |
> | --- | --- | --- | --- | --- |
> | Improvement (%) | 4.20% | 22.35% | 24.62% | 4.05% |
>
> **Iteration Number of Self-Reflection (%):**
>
> | Dataset | 1 | 2 | 3 | 4 | 5 | 5+ |
> | --- | --- | --- | --- | --- | --- | --- |
> | HellaSwag (llama3-70b) | 89.38 | 10.26 | 0.37 | 0.00 | 0.00 | 0.00 |
> | HellaSwag (Claude-3) | 74.54 | 15.74 | 6.48 | 2.31 | 0.93 | 0.00 |
> | TruthfulQA (Llama3-70b) | 88.76 | 6.74 | 2.81 | 0.56 | 0.00 | 1.12 |
> | TruthfulQA (Claude-3) | 81.07 | 16.02 | 1.46 | 0.97 | 0.49 | 0.00 |
> | GSM8K (Claude-3) | 96.94 | 3.06 | 0.00 | 0.00 | 0.00 | 0.00 |
> | GSM8K (Llama3-70b) | 68.45 | 24.40 | 6.55 | 0.60 | 0.00 | 0.00 |
> | MMLU (Claude-3) | 90.29 | 8.86 | 0.86 | 0.00 | 0.00 | 0.00 |
> | MMLU (Llama3-70b) | 71.57 | 15.74 | 7.36 | 4.31 | 0.76 | 0.25 |
>
> From the above results, we can observe the following:
>
> 1. **The DataGen framework exhibits strong generalizability**. Regardless of whether the model is open-weight or proprietary, DataGen demonstrates excellent performance across various modules.
> 2. **DataGen effectively enhances the truthfulness of data generation for many models**. For example, DataGen significantly reduces errors in mathematical problem generation for Llama3-70b, Claude-3, and Llama3-405b (improvements exceeding 50%). Additionally, DataGen's RAG module effectively mitigates factual errors for Claude-3 and Llama3-70b (correction rates exceeding 20%).
>
> ---
>
> We deeply appreciate your thoughtful feedback and valuable suggestions, which have provided insights to refine our work further.  Thanks a lot!

---

> > ### Comment · Reviewer_WuNv · 2024-11-27
> >
> > Thanks for the response. Some of my concerns were addressed, but other reviewers raised others I agree with. I have changed my score to a 6.

---

> ### Author Response · Authors · 2024-11-27
>
> Thank you for your thoughtful feedback. If there are specific aspects of your concern that remain unresolved, we would greatly appreciate it if you could elaborate further so we can address them more effectively.

---

### Author Response · Authors · 2024-11-20
**Thanks for all ACs' and Reviewers' efforts**

First, we would like to thank the Area Chair (AC) and all the reviewers for their time and thoughtful feedback. In response to the reviewers' concerns, we have taken the following measures, which are summarized below:

1. **More Comprehensive Ablation Studies (Reviewer WuNv and Reviewer eQ9J)**: We have added ablation results for Claude-3, Llama3-70B, and Llama3.1-405B on code-based mathematical validation, diversity measurement, the RAG module, and the number of iterations in self-reflection. We have also conducted thorough analyses of these results to demonstrate the generalizability of DataGen.

2. **Inclusion of Additional LLMs in Benchmarking (Reviewer S9ep)**: We have included benchmark results for Llama3.1-405B and provided additional results for Llama3-70B. These results highlight the strong performance of DataGen in dynamic benchmarks.

3. **Justification for Metric Selection (Reviewer 1kEc)**: We have provided a detailed explanation of the calculation process for the remote-clique score and clarified why this metric was chosen to measure diversity. Additional experiment results on alternative metrics (APS and INGF) have also been included in our response.

4. **Effectiveness of RAG (Reviewer S9ep)**: To address the effectiveness of the RAG module in our work, we have presented results for GPT-4, Claude-3, Llama3-70B, and Llama3-405B, along with detailed explanations. Additionally, we have proposed feasible and easily implementable alternatives within our toolkit.

5. **Analysis of GSM8K Benchmark Results (Reviewer S9ep)**: We have incorporated the latest references to support our analysis of GSM8K benchmark results.

6. **Generated Dataset Length Distribution (Reviewer eQ9J)**: We conducted additional experiments to demonstrate that the length of generated datasets can be controlled within DataGen.

7. **Clarifications and Explanations (Reviewers WuNv, 1kEc, and eQ9J)**: We have provided clarifications on data formats, the difficulty enhancements module, and the consistency of prompts used in our experiments.

We hope that our responses address the reviewers’ concerns and further strengthen the understanding of our work. Once again, we sincerely appreciate the reviewers' valuable feedback and efforts.

---

### Meta-Review · Area_Chair_fZqW · 2024-12-20

**Metareview:**

[Summary]
The paper introduces DATAGEN, a unified framework leveraging large language models (LLMs) to generate diverse, accurate, and controllable textual datasets. The framework incorporates innovative modules such as attribute-guided generation, code-based label verification, and retrieval-augmented generation to address challenges in generalization, diversity, truthfulness, and user-defined constraints.

[Strengths]
  - Compared to related work, DATAGEN demonstrates the ability to cover a wide range of features in real-world scenarios.
  - The framework is well-motivated, with a simple yet generalizable design that can be applied across various domains and setups.

[Weaknesses]
  - The ablation study and module analysis are somewhat simplified, limiting the validation of each module's effectiveness.
  - The paper contains some ambiguities, particularly in areas such as data formatting, diversity metrics, length distribution of generated data, and difficulty enhancements.

[Decision]
This paper is well-motivated and presents a comprehensive framework that effectively addresses the challenges of dataset generation using LLMs. Based on the reviewers’ recommendations (6: WuNv, 6: 1kEc, 6: S9ep, 6: eQ9J), I recommend accepting this paper.

**Additional Comments On Reviewer Discussion:**

During the rebuttal period, the reviewers provided helpful feedback and clarifying questions. The authors addressed most of these concerns effectively.
  - Lack of Ablation Studies: To address the lack of ablation studies, the authors added ablation results for Claude-3, Llama3-70B, and Llama3.1-405B on each module.
  - Ambiguities in Writing: To address ambiguities, the authors clarified by providing additional experiments and explanations

---

### Decision · Program_Chairs · 2025-01-22

Accept (Poster)